# PaT: Planning-after-Trial for Efficient Code Generation

## Abstract

Large language models (LLMs) have demonstrated increasingly sophisticated capabilities for code generation. To extend the problem-solving reach of cost-efficient models to complex problems, strategic planning via problem decomposition has emerged as a key paradigm. However, most existing pipelines adopt a rigid Planning-before-Trial (PbT) policy, which inefficiently allocates test-time compute by incurring planning overhead even on directly solvable problems. We propose an adaptive Planning-after-Trial (PaT) policy that uses the outcome of a direct attempt as a feedback signal, invoking a planner only upon verification failure. This adaptive policy naturally enables a heterogeneous model configuration: a cost-efficient model handles generation attempts, while a powerful model is reserved for targeted planning interventions. Empirically, across multiple benchmarks and model families, our approach significantly advances the cost-accuracy Pareto frontier by judiciously avoiding indiscriminate planning on simple problems and concentrating test-time compute precisely where it is needed most.

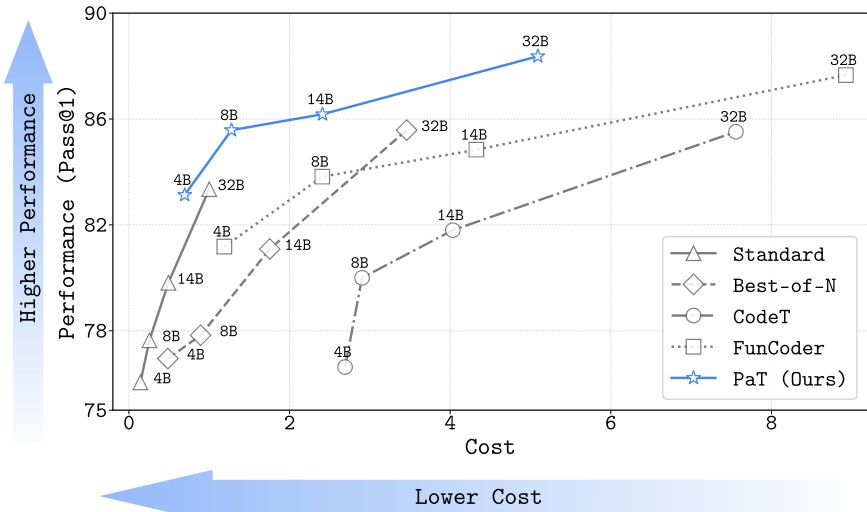

Figure 1: **Cost-performance trade-off across different code generation strategies.** Proposed PaT policy consistently outperforms all baselines, establishing a state-of-the-art Pareto frontier.

## 1 Introduction

The code generation capabilities of large language models (LLMs) have advanced significantly with model and data scaling (Chen et al., 2021; Li et al., 2022; OpenAI, 2023), spurring their integration into practical developer tools (Schick et al., 2023). However, widespread adoption is hindered by high inference cost and latency. This challenge is particularly acute because real-world workloads are composed of problems with widely varying intrinsic difficulty (Khan et al., 2023): some are simple enough for a direct solution, while others are complex and require sophisticated planning. Applying a uniform, high computational budget to this distribution is fundamentally inefficient and difficult to operate at scale.

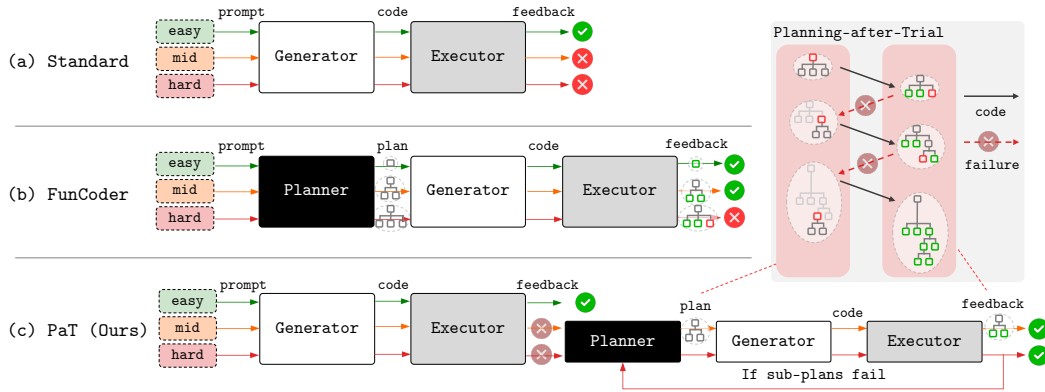

Figure 2: **Comparison with existing methods and PaT (Ours).** Problems are grouped by difficulty (easy, mid, and hard). Boxes denote the key components: a Generator (creates code), a Planner (decomposes the problem), and an Executor (verifies the solution). **(a) Standard** directly generate and execute; works on easy problems but often fails on harder ones. **(b) FunCoder (PbT)** always plans first, so planning cost is paid even when unnecessary. **(c) PaT (Ours)** trials first and plans only on failure; solves easy problems cheaply and hard problems adaptively.

A common approach to balance cost and performance is to employ a cost-efficient language model. While this strategy is effective for simple implementations, it inevitably hits a capability ceiling on more complex problems, sacrificing performance. To address this limitation, a natural approach is to decompose a complex coding problem into a plan of simpler helper functions (Zhou et al., 2023; Jiang et al., 2024b; Chen et al., 2024). However, these methods typically adopt a rigid Planning-before-Trial (PbT) policy, where decomposition occurs before any solution trial is made. As illustrated in Figure 2, this pre-emptive planning is fundamentally inefficient. It incurs unnecessary decomposition overhead on problems that could have been solved directly. Furthermore, the decomposition plan is often created without accounting for the solver model's actual capability, potentially yielding helper functions that remain too difficult to implement. Consequently, such PbT policy is often both inefficient on simple problems and ineffective on complex ones.

To achieve both efficiency on simple problems and effectiveness on complex ones, we introduce Planning-after-Trial (PaT), an adaptive policy that inverts the rigid PbT policy. PaT first attempts a direct solution and verifies it; only when that attempt fails does it trigger the decomposition. As illustrated in Figure 2, PaT ensures that planning is invoked only for problems that demonstrably require it by using feedback of a direct attempt. Consequently, as validated by our results in Figure 1, PaT achieves superior performance while significantly lowering the average computational cost, thereby advancing the cost-performance Pareto frontier. This efficiency stems from keeping simple problems cheap and concentrating resources precisely on hard ones.

The efficiency of the PaT policy is further amplified within a heterogeneous model configuration. This enables a strategic division of labor, where a cost-efficient model is assigned to the high-volume 'generator' role and a powerful one to the infrequent 'planner' role. This setup is uniquely synergistic with PaT; while a rigid PbT policy would invariably incur the high cost of the powerful planner on every problem, PaT reserves this expensive resource exclusively for the instances where the generator fails. We provide a formal analysis of this heterogeneous approach, establishing the conditions for its superior efficiency.

We empirically validate our claims through a comprehensive evaluation across diverse model families and standard benchmarks, evaluated with the robust EvalPlus (Liu et al., 2023) test suite. Our results first establish that the PaT policy consistently outperforms the state-of-the-art PbT baseline, FunCoder (Chen et al., 2024), achieving higher performance across all model scales while using, on average, only 60% of its computational resources. We then show that our proposed heterogeneous configuration further enhances this efficiency. For instance, a small generator model guided by a powerful planner achieves performance within 1% of a homogeneous powerful model, while operating at only 31% of the cost. These findings confirm that PaT's adaptive policy establishes a new, superior cost-performance frontier.

Our key contributions are summarized as follows:

- We propose Planning-after-Trial (PaT), an adaptive policy that invokes planning only upon verification failure to avoid unnecessary overhead.
- We pair the PaT policy with a heterogeneous model configuration, combining small models for generation and large models for planning, and provide a theoretical analysis of its cost-efficiency.
- We provide a comprehensive evaluation across model families and benchmarks, showing that our approach consistently advances the cost-performance Pareto frontier.

## 2 RELATED WORK

**Code Generation with LLMs.** A variety of methods have been explored to enhance the effectiveness of LLMs for code generation. Foundational techniques focus on improving the initial output, such as few-shot prompting (Brown et al., 2020) or sampling multiple candidates with Best-of-N to leverage diversity (Chen et al., 2021). To verify and refine these candidates, test-driven approaches like CodeT (Chen et al., 2022a) and self-debugging methods (Shinn et al., 2023; Zhong et al., 2024) leverage feedback loops. For more complex tasks, recent systems introduce hierarchical decomposition (Chen et al., 2024; Le et al., 2024), utilize natural language planning to guide search (Wang et al., 2025a), or employ interactive chains of repair (Wang et al., 2024; Zhang et al., 2024). Another line of work enriches the model's input by retrieving relevant code snippets or API documentation from external knowledge bases (Zhang et al., 2023; Li et al., 2022). Most notably, AdaCoder (Zhu et al., 2025) share our switching concept but treat planning as iterative refinement, a flat approach limited on hard problems. In contrast, PaT employs hierarchical decomposition, tackling harder problems by breaking them down into sub-problems.

**Divide-and-conquer strategies for complex reasoning.** To tackle complex problems beyond the reach of direct generation, divide-and-conquer for LLMs has evolved from implicit decomposition through structured reasoning to more explicit, structured planning. Early forms include the linear step-by-step process in Chain-of-Thought (CoT) (Wei et al., 2022) and the exploration of multiple reasoning paths in Tree-of-Thoughts (ToT) (Yao et al., 2023). More recent methods create an explicit plan of sub-problems before implementation across various domains, from solving problems sequentially in Least-to-Most prompting (Zhou et al., 2023), to mathematical reasoning with Program of Thoughts (Chen et al., 2022b), using a society of models for complex reasoning (Juneja et al., 2024) and code generation with structured pipelines like Self-Plan (Jiang et al., 2024b) and FunCoder (Chen et al., 2024). Despite their variations, these approaches largely adhere to Planning-before-Trial (PbT). Recent work has recognized the inefficiency of this rigid approach, leading to methods that attempt to learn an adaptive policy for when to invoke planning (Paglieri et al., 2025). In contrast, our PaT policy offers a simpler solution: it makes decomposition a reactive intervention triggered by a direct, learning-free feedback signal, avoiding the universal cost of PbT and the complexity of training a separate policy model.

## 3 PAT: PLANNING AFTER TRIAL

**Problem Formulation.** We model a code generation instance as a specification $x$ (e.g., natural language description with I/O signature) and seek a program $f$ that satisfies $x$. Let $M_G$ denote a generator model and $M_P$ a planner model. For any specification $x$, the generator $M_G$ produces a direct candidate implementation, denoted $\hat{f}$. The planner $M_P$, when invoked, produces a decomposition plan consisting of a new top-level implementation, $\hat{f}$, and a set of subproblem specification $\{x_i\}$. The final program $\mathcal{F}$ is constructed by COMPOSE that merges an implementation $\hat{f}$ with a set of verified helper functions $H$. For verification, we construct a test set $\mathcal{T}(x) = \{(\text{in}_j, \text{out}_j)\}_{j=1}^t$ and execute programs in a sandboxed Python runtime. We evaluate $\mathcal{F}$ with

$$\text{EVALUATE}\big(\mathcal{F}, \mathcal{T}(x)\big) = \sum_{j=1}^{t} \mathbf{1}\big[\mathcal{F}(\text{in}_j) = \text{out}_j\big], \tag{1}$$

*i.e.,* the number of tests passed.

**Algorithm 1** PAT

1: **Input:** Problem $x$,
2:         Helper Functions $H$,
3:         Generator Model $M_G$,
4:         Planner Model $M_P$
5: **Output:** Generated program $\mathcal{F}$
6: $\hat{f} \leftarrow M_G(x; H)$
7: $\mathcal{T} \leftarrow \text{GENERATETESTS}(x)$
8: $p \leftarrow \text{EVALUATE}(\text{COMPOSE}(\hat{f}, H), \mathcal{T})$
9: **if** $p = |\mathcal{T}|$ **then**
10:     $\mathcal{F} \leftarrow \text{COMPOSE}(\hat{f}, H)$
11:     **return** $\mathcal{F}$
12: **end if**
13: $\mathcal{F} \leftarrow \text{DIVIDE}(x, \hat{f}, p, H, \mathcal{T}, M_G, M_P)$
14: **return** $\mathcal{F}$

**Algorithm 2** DIVIDE

1: **Input:** $x, \hat{f}, p, H, \mathcal{T}, M_G, M_P$
2: **Output:** $\mathcal{F}$
3: **while** improvement exists **do**
4:     $\hat{f}_{\text{prev}}, p_{\text{prev}} \leftarrow \hat{f}, p$
5:     $\hat{f}, \{x_i\} \leftarrow M_P(x; H)$
6:     **for** each $x_i$ not implemented in $H$ **do**
7:         $H \leftarrow \text{PAT}(x_i, H, M_G, M_P)$
8:     **end for**
9:     $p \leftarrow \text{EVALUATE}(\text{COMPOSE}(\hat{f}, H), \mathcal{T})$
10:     **if** $p = |\mathcal{T}|$ **then return** $\text{COMPOSE}(\hat{f}, H)$
11:     **else if** $p \leq p_{\text{prev}}$ **then**
12:         **return** $\text{COMPOSE}(\hat{f}_{\text{prev}}, H)$
13:     **end if**
14: **end while**

Figure 3: **Pseudocode for PaT.** $\text{GENERATETESTS}(x)$ constructs the unit-test set $\mathcal{T}$ from the specification. $\text{COMPOSE}(\hat{f}, H)$ merges verified helper implementations into the parent program. The planner is invoked only on failure, and verified subsolutions are reused as context.

## 3.1 ADAPTIVE PLANNING VIA FAILURE FEEDBACK

PaT executes a simple failure-triggered policy. Given problem specification $x$, the generator $M_G$ first produces a monolithic candidate solution $\hat{f} \leftarrow M_G(x)$, which we verify on a test set $\mathcal{T}(x)$. If all tests pass, *i.e.,* $\text{EVALUATE}(\hat{f}, \mathcal{T}(x)) = |\mathcal{T}(x)|$, the pipeline terminates and returns $\hat{f}$ without incurring any decomposition overhead. If verification fails, PaT invokes the planner $M_P$ to produce a set of subproblem specifications $\{x_i\}_{i=1}^{m} \leftarrow M_P(x)$ together with a composition rule for subsolutions. Each $x_i$ is then handled recursively by the same policy: attempt a solution $\hat{f}_i \leftarrow M_G(x_i)$; verify; only on failure, decompose $x_i$ further.

When all subproblems have verified solutions, *i.e.,* $\text{EVALUATE}(\hat{f}_i, \mathcal{T}(x_i)) = |\mathcal{T}(x_i)|$ for all $i$, PaT composes them into a parent-level candidate and re-verifies against $\mathcal{T}(x)$. If this composite solution passes, the process terminates successfully. Otherwise, the planner is invoked again on the original problem specification $x$. For this subsequent planning attempt, the planner is provided with the original problem and the set of previously successful subsolutions $\{\hat{f}_i\}$ as additional context. This allows the planner to make a more informed decomposition decision while reusing already successful components, further enhancing cost-efficiency.

## 3.2 TEST CASES AND VERIFICATION

Relying solely on provided public test cases is often insufficient, as benchmarks like MBPP provide none at all (Austin et al., 2021) while others lack critical edge cases (Chen et al., 2021; Khan et al., 2023). To ensure robust operation in such environments, we explicitly generate comprehensive unit tests $\mathcal{T}(x)$. However, prior works note that generated tests can be noisy or incorrect (Chen et al., 2022a; Wang et al., 2025b; Prasad et al., 2025). To mitigate this, many systems adopt consensus-based scoring to identify the most robust solution (Chen et al., 2022a; 2024). This approach evaluates a pool of candidate solutions against numerous tests, selecting the one that achieves the broadest consensus. However, such consensus-based scoring is ill-suited for our framework, as its objective of selecting the most likely correct solution from a pool contrasts with PaT's need for a definitive binary signal (success or failure) to trigger its adaptive escalation to planning.

We therefore separate the success criterion from the termination rule. The success criterion is strict: a candidate is accepted only if it passes all tests in $\mathcal{T}(x)$. To handle noisy tests and prevent unproductive recursion, we track the pass count $p^{(t)} = \text{EVALUATE}(\hat{f}, \mathcal{T}(x))$ at iteration $t$ and apply a plateau rule: if $p^{(t)} \leq p^{(t-1)}$ then halt and return the best-known solution. Intuitively, if the pass

count does not strictly improve, we assume all valid tests have been satisfied and remaining failures are due to flawed tests. To avoid overfitting to potentially incorrect failing tests, we do not provide those failing cases as explicit feedback to the generator.

# 4 HETEROGENEOUS MODEL CONFIGURATION FOR FURTHER EFFICIENCY

To further enhance the efficiency of PaT, we adopt a heterogeneous model configuration that assigns different models to distinct roles. PaT naturally decomposes into two roles: a **Generator** that attempts to solve the current specification, and a **Planner** that decomposes the problem upon failure. These roles place different demands on model capability. The Generator's task, which is to produce a solution for a well-scoped and often manageable subproblem, can be handled effectively by a cost-efficient small LM (sLM). In contrast, the Planner's task, which is to understand the nuances of a complex specification and propose a helpful decomposition, benefits from the advanced reasoning of a high-performance large language model (LLM).

Building on this observation, we instantiate PaT with an sLM as the Generator and an LLM as the Planner. However, efficiency is not guaranteed by "using a smaller model" alone. Decomposition, while cheaper than full program generation, is not free; it consumes planning tokens. If the sLM is too weak, failures become frequent, triggering the (more expensive) Planner often and raising total cost. If the sLM is too strong, its per-call cost approaches that of the LLM, shrinking the benefit of heterogeneity. Thus, there is a critical trade-off: a very cheap but under-capable sLM increases the number of planning interventions, whereas a stronger sLM reduces interventions but erodes cost savings. In Section 4.1 we show that, for a fixed Planner (LLM) and under a sufficient overhead condition, there exists a Generator capability (sLM) for which heterogeneous PaT is more cost-efficient in expectation than an LLM-only policy.

## 4.1 THEORETICAL ANALYSIS FOR THE HETEROGENEOUS MODEL CONFIGURATION

This section analyzes when PaT with a heterogeneous model configuration, using a small model $s$ for generation and a large model $L$ for planning, achieves lower expected cost than an LLM-only policy. Problem complexity is denoted by $k > 0$; for $M \in \{L, s\}$, write $p_M$ for the problem solving capability and $c_M$ for the unit cost, with $0 < p_s < p_L$ and $0 < c_s < c_L$.

**Assumption 1** (Generation cost). *A generation call by $M_G$ on a problem of complexity $k$ incurs*

$$Cost_{Generation} = \begin{cases} k\,c_{M_G}, & k \le p_{M_G} \quad \text{(success)}, \\ p_{M_G}\,c_{M_G}, & k > p_{M_G} \quad \text{(failed attempt)}. \end{cases} \tag{2}$$

**Assumption 2** (Planning cost). *A Planner-produced $n$-way plan maps a problem of size $k$ to $n$ independent subproblems of size $\frac{k}{n}$ and incurs a fixed per-subproblem overhead $D_{M_P}$, so that*

$$Cost_{Planning} = n D_{M_P}. \tag{3}$$

Planning introduces a fixed overhead, so an overly weak Generator triggers frequent planning while an overly strong one narrows the cost gap to the LLM. Efficiency therefore hinges on selecting an appropriately capable sLM. To guide this choice, Theorem 1 gives a sufficient condition guaranteeing the existence of a Generator capability $p_s \in (0, p_L)$ under which PaT with a Heterogeneous Model Configuration is strictly cheaper in expectation than an LLM-only policy.

**Theorem 1** (Existence of an efficient sLM capability). *Let $k \sim \mathrm{Unif}(0, p_L]$. Under Assumptions 1 and 2, if the total planning overhead satisfies*

$$n D_L \;<\; \left( \tfrac{1}{2} - \tfrac{1}{n^2} \right) p_L c_L, \tag{4}$$

*then there exists $p_s \in (0, p_L)$ for which PaT with a Heterogeneous Model Configuration has strictly lower expected cost than an LLM-only policy.*

We assume $k \sim \mathrm{Unif}(0, p_L]$, which LLM can solve in one call while sLM needs some decomposition. The term $p_L c_L$ serves as a scale for a one-shot LLM solve (capability times per-unit cost). Intuitively, the inequality says planning must be sufficiently cheap relative to an LLM call; under this condition there exists an sLM capability $p_s$ that makes PaT with a Heterogeneous Model Configuration strictly cheaper in expectation than LLM-only. Full proof and further asymptotic analysis for larger $k$ are given in Appendix A.

## 5 EXPERIMENTS

In this section, we evaluate PaT in terms of both performance and cost efficiency. We conduct experiments on two settings: a homogeneous setting (Section 5.1), and a heterogeneous setting (Section 5.2). Below we describe the experimental setup used in our evaluation, including benchmarks, LLMs, evaluation metrics, and baselines.

**Benchmarks.**   We evaluate on established code generation benchmarks. **HumanEval** (Chen et al., 2021) and **MBPP** (Austin et al., 2021) measure foundational code generation. For both, we use EvalPlus (Liu et al., 2023) to obtain expanded and more robust unit tests (*i.e.*, **HumanEval+** and **MBPP+**). For complex tasks, we use **xCodeEval** (Khan et al., 2023) benchmark, partitioning instances into four categories (Easy, Mid, Hard, and Expert) following FunCoder's rating scheme.

**LLMs.**   We primarily adopt the Qwen3 family (4B, 8B, 14B, and 32B) (Yang et al., 2025) as our main testbed, since it provides fine-grained scaling steps within a single model family. To evaluate the cross-family generalization of our method, we additionally conduct experiments on Llama-3.1-8B-Instruct (Dubey et al., 2024) and DeepSeek-Coder-V2-Lite-Instruct ($\approx$16B) (Zhu et al., 2024). Table 1 reports input and output token prices (USD per million tokens) collected from public provider listings[1]. Since there is no official price available for DeepSeek-Coder-V2-Lite-Instruct, we use the pricing policy of DeepSeek-Coder-V2 as a proxy.

Table 1: **Token pricing (USD per million tokens).** The values represent the cost for input and output, respectively.

| Model | Input | Output |
|---|---|---|
| Qwen3$_{4B}$ | 0.11 | 0.42 |
| Qwen3$_{8B}$ | 0.18 | 0.70 |
| Qwen3$_{14B}$ | 0.35 | 1.40 |
| Qwen3$_{32B}$ | 0.70 | 2.80 |
| Llama3.1$_{8B}$ | 0.10 | 0.10 |
| DeepSeek-Coder | 0.14 | 0.28 |

**Evaluation Metric.**   We report two primary metrics: **Pass@1** and **LLM cost**. Pass@1 is our primary performance metric, reflecting the ability to generate a correct solution on the first attempt, which aligns with real-world efficiency goals. LLM cost for each experiment is computed directly from the per-token prices in Table 1.

**Baselines.**   To validate the effectiveness of PaT, we compare it against the following state-of-the-art code generation methods:

- **Standard** (Brown et al., 2020) uses few-shot prompting, giving several in-context examples to the model and asking it to generate a solution; we follow the original prompting protocol to measure baseline quality without post-hoc filtering or decomposition.

- **Best-of-N** (Chen et al., 2021) generates multiple candidate programs per problem (we use N=5) and selects the best-performing one by executing candidates against the test suite, emphasizing sampling diversity and selection to boost pass@k.

- **CodeT** (Chen et al., 2022a) performs iterative, test-driven refinement by automatically generating additional unit tests, running generated solutions against these tests, and using failing cases to guide subsequent generations for improved robustness.

- **FunCoder** (Chen et al., 2024) implements a two-stage static divide-and-conquer pipeline that decomposes tasks according to a fixed hierarchy, solves subproblems, and assembles the final solution; its decomposition plan is precomputed and does not adapt to generation feedback.

For the sake of reproducibility and clarity, a detailed account of our implementation is included in Appendix B-F.

### 5.1 HOMOGENEOUS SETTING

In this section, we compare results from running PaT using a homogeneous language model against baselines. Evaluations are performed on the foundational benchmarks (HumanEval and MBPP; Sec. 5.1.1) and the complex benchmark (xCodeEval; Sec. 5.1.2).

---

[1] https://artificialanalysis.ai (accessed 2025-09-25).

Table 2: **Performance and cost comparison on foundational benchmarks.** We report Pass@1 across four benchmarks (HumanEval, HumanEval+, MBPP, and MBPP+) and their average ('Avg.'). Cost is normalized relative to the Standard baseline (1.00). Best results are in **bold**.

| Model | Method | HumanEval | | MBPP | | Avg. | Δ Avg. | Cost |
|---|---|---|---|---|---|---|---|---|
| | | base | plus | base | plus | | | |
| Qwen3$_{4B}$ | Standard | 78.66 | 70.12 | 79.43 | 76.00 | 76.05 | - | 1.00 |
| | Best-of-N | 79.27 | 71.95 | 80.00 | 76.57 | 76.95 | + 0.90 | 3.39 |
| | CodeT | 78.66 | 70.73 | 80.00 | 77.14 | 76.63 | + 0.58 | 18.82 |
| | FunCoder | 85.98 | 79.88 | 81.14 | 77.71 | 81.18 | + 5.13 | 8.31 |
| | PaT (Ours) | **89.63** | **82.32** | **82.29** | **78.29** | **83.13** | + 7.08 | 4.85 |
| Qwen3$_{8B}$ | Standard | 78.66 | 71.34 | 81.14 | 79.43 | 77.64 | - | 1.00 |
| | Best-of-N | 80.49 | 72.56 | 80.57 | 77.71 | 77.83 | + 0.19 | 3.50 |
| | CodeT | 80.49 | 73.78 | 84.57 | 81.14 | 80.00 | + 2.36 | 11.37 |
| | FunCoder | 89.02 | 81.10 | 83.43 | 81.71 | 83.82 | + 6.18 | 9.43 |
| | PaT (Ours) | **90.85** | **82.32** | **85.14** | **84.00** | **85.58** | + 7.94 | 5.00 |
| Qwen3$_{14B}$ | Standard | 83.54 | 76.83 | 80.57 | 78.29 | 79.81 | - | 1.00 |
| | Best-of-N | 82.93 | 77.44 | 82.86 | 81.14 | 81.09 | + 1.28 | 3.58 |
| | CodeT | 83.54 | 76.22 | **85.71** | 81.71 | 81.80 | + 1.99 | 8.22 |
| | FunCoder | 89.63 | 81.71 | 85.14 | 82.86 | 84.84 | + 5.03 | 8.82 |
| | PaT (Ours) | **91.46** | **83.54** | **85.71** | **84.00** | **86.18** | + 6.37 | 4.91 |
| Qwen3$_{32B}$ | Standard | 87.20 | 80.49 | 83.43 | 82.29 | 83.35 | - | 1.00 |
| | Best-of-N | 90.24 | 82.93 | 85.14 | 84.00 | 85.58 | + 2.23 | 3.46 |
| | CodeT | 89.02 | 80.49 | 86.86 | 85.71 | 85.52 | + 2.17 | 7.56 |
| | FunCoder | 93.29 | **84.76** | 86.86 | 85.71 | 87.66 | + 4.31 | 8.93 |
| | PaT (Ours) | **93.90** | 84.15 | **88.57** | **86.86** | **88.37** | + 5.02 | 5.09 |
| Llama3.1$_{8B}$ | Standard | 68.29 | 59.15 | 66.86 | 66.29 | 65.15 | - | 1.00 |
| | Best-of-N | 70.12 | 62.20 | 72.57 | 69.14 | 68.51 | + 3.36 | 2.60 |
| | CodeT | 71.95 | 61.59 | 72.00 | 71.43 | 69.24 | + 4.09 | 4.86 |
| | FunCoder | 75.00 | 67.68 | 72.00 | 71.43 | 71.53 | + 6.38 | 8.07 |
| | PaT (Ours) | **78.05** | **68.90** | **74.29** | **72.00** | **73.31** | + 8.16 | 5.32 |
| DeepSeek-Coder | Standard | 83.54 | 75.61 | 80.57 | 78.86 | 79.65 | - | 1.00 |
| | Best-of-N | 81.10 | 75.00 | 81.71 | 81.14 | 79.74 | + 0.09 | 3.05 |
| | CodeT | 81.10 | 72.56 | 84.57 | 82.86 | 80.27 | + 0.62 | 6.40 |
| | FunCoder | 85.98 | 79.27 | **85.14** | 84.00 | 83.60 | + 3.95 | 8.77 |
| | PaT (Ours) | **86.59** | **79.88** | **85.14** | **85.14** | **84.19** | + 4.54 | 5.97 |

### 5.1.1 FOUNDATIONAL BENCHMARKS (HUMANEVAL AND MBPP)

We first evaluate PaT in a homogeneous setting on the foundational HumanEval and MBPP benchmarks, including their extended EvalPlus versions. As shown in Table 2, PaT consistently outperforms all baselines across all tested model families and scales. The effectiveness of this policy is best illustrated by its ability to dramatically improve the capability of smaller models. For instance, PaT enables the Qwen3$_{4B}$ model to achieve an average Pass@1 of 83.13%, which is remarkably similar to the 83.35% achieved by Standard on Qwen3$_{32B}$ model, a model eight times larger. This demonstrates that PaT is not merely an incremental improvement but a powerful policy that fundamentally alters the performance curve of a given model.

Beyond its superior performance, PaT also shows critical advantages in cost-efficiency. As detailed in Table 2, PaT consistently achieves higher performance than FunCoder, the previous state-of-the-art hierarchical method, while consuming, on average 60% of its cost. This efficiency gain can be explained by examining Standard performance. On these foundational benchmarks, Standard solves, on average, 76% of problems directly, demonstrating that a large majority of tasks do not require decomposition. FunCoder's rigid PbT policy is forced to pay a cost of planning on all of these problems, incurring its expensive planning overhead even when it is not needed. In contrast, PaT only invokes its planner on the much smaller fraction of problems (24%) that actually fail the initial, cheaper trial. By avoiding this universal planning overhead, PaT establishes a new and more effective cost-performance frontier for code generation.

Table 3: **Performance breakdown on xCodeEval by difficulty.** We report Pass@1 scores for each difficulty category (Easy, Mid, Hard, and Expert) and the overall score.

| Model | Method | Easy | Mid | Hard | Expert | All | $\Delta$ All | Cost |
|---|---|---|---|---|---|---|---|---|
| Qwen3_{4B} | Standard | 37.70 | 17.86 | 3.45 | 0.00 | 18.40 | - | 1.00 |
| | Best-of-N | 51.91 | 29.46 | 8.05 | 0.00 | 27.00 | + 8.60 | 4.06 |
| | CodeT | 40.44 | 20.54 | 3.45 | 0.00 | 20.00 | + 1.60 | 21.64 |
| | FunCoder | 55.19 | 29.46 | 12.64 | 0.00 | 29.00 | + 10.60 | 12.95 |
| | PaT (Ours) | **61.75** | **40.18** | **14.94** | 0.00 | **34.20** | + 16.20 | 17.93 |
| Qwen3_{8B} | Standard | 54.10 | 28.57 | 5.75 | 0.00 | 27.20 | - | 1.00 |
| | Best-of-N | 66.67 | 42.86 | 6.90 | 0.00 | 35.20 | + 8.00 | 3.34 |
| | CodeT | 45.92 | 31.25 | 8.05 | 0.00 | 25.20 | - 2.00 | 17.00 |
| | FunCoder | 64.48 | 43.75 | 9.20 | 0.00 | 35.00 | + 7.80 | 8.62 |
| | PaT (Ours) | **69.95** | **45.54** | **11.49** | 0.00 | **37.80** | + 10.60 | 6.98 |
| Qwen3_{14B} | Standard | 53.55 | 36.61 | 9.20 | 0.00 | 25.20 | - | 1.00 |
| | Best-of-N | 68.31 | 50.00 | 13.79 | 0.00 | 38.60 | + 13.40 | 3.16 |
| | CodeT | 55.19 | 41.07 | 9.20 | 0.00 | 31.00 | + 5.80 | 7.48 |
| | FunCoder | 73.22 | 52.68 | 18.39 | 0.00 | 41.80 | + 16.60 | 9.03 |
| | PaT (Ours) | **73.77** | **53.57** | **21.84** | **0.85** | **43.00** | + 17.80 | 6.49 |
| Qwen3_{32B} | Standard | 54.64 | 39.29 | 11.49 | 0.00 | 30.80 | - | 1.00 |
| | Best-of-N | 71.04 | 50.00 | 14.94 | 0.00 | 39.80 | + 9.00 | 3.28 |
| | CodeT | 57.92 | 39.29 | 12.64 | 0.00 | 32.20 | + 1.40 | 7.55 |
| | FunCoder | **74.86** | **54.46** | 16.09 | 0.00 | 42.40 | + 11.60 | 7.87 |
| | PaT (Ours) | 74.32 | **54.46** | **18.39** | **1.69** | **43.00** | + 12.20 | 6.00 |
| Llama3.1_{8B} | Standard | 13.11 | 3.57 | 1.15 | 0.00 | 5.80 | - | 1.00 |
| | Best-of-N | 22.40 | 8.04 | 1.15 | 0.00 | 10.20 | + 4.40 | 1.39 |
| | CodeT | 13.11 | 3.57 | 1.15 | 0.00 | 5.80 | + 0.00 | 2.59 |
| | FunCoder | 26.78 | 7.14 | **2.30** | 0.00 | 11.80 | + 6.00 | 5.46 |
| | PaT (Ours) | **32.79** | **11.61** | **2.30** | 0.00 | **15.00** | + 14.20 | 8.42 |
| DeepSeek-Coder | Standard | 43.17 | 19.64 | 9.20 | 0.85 | 22.00 | - | 1.00 |
| | Best-of-N | 59.56 | 26.79 | 10.34 | 0.85 | 29.80 | + 7.80 | 2.57 |
| | CodeT | 46.99 | 22.32 | 9.20 | 0.00 | 23.80 | + 1.80 | 5.46 |
| | FunCoder | 62.30 | 31.25 | 12.64 | **1.69** | 32.40 | + 10.40 | 8.69 |
| | PaT (Ours) | **63.39** | **33.93** | **13.79** | **1.69** | **33.60** | + 11.60 | 7.69 |

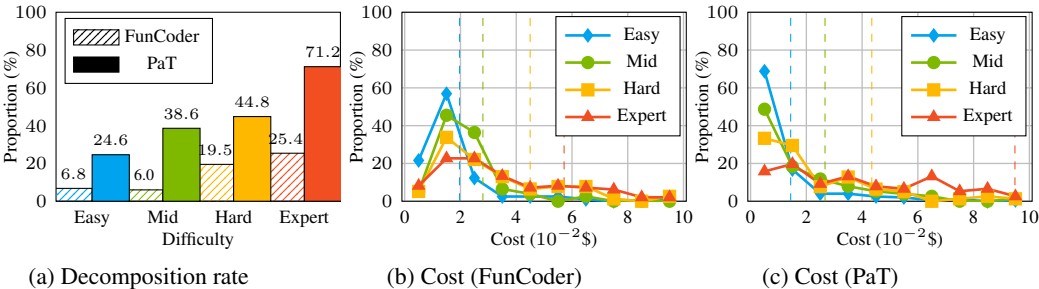

(a) Decomposition rate      (b) Cost (FunCoder)      (c) Cost (PaT)

Figure 4: **Adaptive decomposition probability and cost analysis for Qwen3_{4B} on xCodeEval.** (a) Decomposition rate of FunCoder and PaT by problem difficulty. Per-difficulty cost distribution (solid line) and average cost (vertical dashed line) on (b) FunCoder and (c) PaT.

### 5.1.2 COMPLEX BENCHMARK (XCODEEVAL)

We evaluate PaT and baselines on the more challenging xCodeEval benchmark. On this benchmark, PaT again achieves significantly higher performance than all baselines across all model scales. However, we observe an interesting cost dynamic: for smaller models like Qwen3_{4B} and Llama3.1_{8B}, PaT incurs a higher cost than FunCoder. This is not a sign of inefficiency but a direct consequence of PaT's adaptive strategy. Less capable models fail more frequently on xCodeEval's difficult problems, and PaT correctly interprets these failures as signals to invest in decomposition. This strategic escalation, while costly, is precisely what allows these smaller models to overcome complex challenges where rigid policies like FunCoder's fall short.

| Generator | Planner | Avg. | Cost |
|-----------|---------|------|------|
| Qwen3$_{32B}$ | Qwen3$_{32B}$ | 88.37 | 1.00 |
| Qwen3$_{14B}$ | Qwen3$_{14B}$ | 86.18 | 0.47 |
|  | Qwen3$_{32B}$ | 87.53 | 0.49 |
| Qwen3$_{8B}$ | Qwen3$_{8B}$ | 85.58 | 0.25 |
|  | Qwen3$_{32B}$ | 87.39 | 0.31 |
| Qwen3$_{4B}$ | Qwen3$_{4B}$ | 83.13 | 0.14 |
|  | Qwen3$_{32B}$ | 84.78 | 0.18 |

(a) Performance (Pass@1) and relative cost results

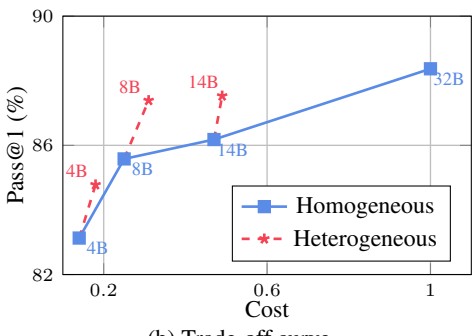

(b) Trade-off curve

Figure 5: **Heterogeneous configurations achieve superior cost-effectiveness.** (a) Performance and relative cost on four code generation benchmarks (HumanEval, HumanEval+, MBPP, and MBPP+). The cost is normalized relative to the homogeneous Qwen3$_{32B}$ model, which is set to 1.00. Full results are provided in Appendix G. (b) The trade-off curve visualizes these results.

For a deeper analysis, we examine the behavior of the Qwen3$_{4B}$ model in Figure 4, which reveals a fundamental limitation of pre-emptive planning. While both PaT and FunCoder increase decomposition for harder problems, PaT exhibits a much more dynamic response compared to FunCoder's more restrained increase. On 'Easy' problems, PaT's decomposition rate (24.6%) appears higher than FunCoder's (6.8%), yet its cost is lower. This is because FunCoder, as a rigid PbT policy, incurs a planning cost on 100% of problems, regardless of whether it ultimately decomposes them. In contrast, PaT only pays this cost for the fraction of problems that fail an initial trial. This validates a core principle of our approach: planning is an expensive operation that must be invoked judiciously, a strategic decision that is only possible with the feedback from an initial trial.

### 5.2 HETEROGENEOUS SETTING

Our theoretical analysis in Section 4 established the potential for enhanced cost-efficiency in a heterogeneous model configuration. To empirically validate this, we conduct experiments on the foundational benchmarks, pairing a powerful planner (Qwen3$_{32B}$) with a series of smaller generator models. The results, presented in Table 5a, provide strong support for our analysis. Pairing a Qwen3$_{8B}$ generator with the Qwen3$_{32B}$ planner achieves an average Pass@1 of 87.39%, coming within 1% of the performance of the homogeneous Qwen3$_{32B}$, while reducing the relative cost to just 0.31.

The superior cost-benefit trade-off of the heterogeneous approach is visualized in Figure 5b. In this graph, the slope of the curve represents the performance return on additional cost. The heterogeneous model configurations exhibit a significantly steeper slope, demonstrating that upgrading only the planner is a highly capital-efficient strategy, yielding substantial performance gains for a marginal increase in cost. This confirms our central hypothesis: because PaT invokes the planner infrequently, reserving a powerful model for this critical but rare task is the most cost-effective way to enhance the overall system's capability.

### 5.3 ANALYSIS OF GENERATED TEST CASES

To assess the reliability of our self-verification mechanism, we quantitatively analyze generated test cases on HumanEval. PaT generates an average of 6.7 test cases per problem, providing significantly broader coverage than standard public examples. Crucially, we examine the distribution of false positives, defined as incorrect test cases that fail valid solutions, to determine if verification noise is systemic. As illustrated in Figure 6, the results confirm that the verification signal is robust for the vast majority of tasks. For instance, with Qwen3$_{4B}$, 63.4% of test cases are completely error-free. Furthermore, instances of severe noise (3+ false positives) are concentrated in a small minority of problems. In these specific noisy instances, the pass rate inevitably saturates because the model cannot satisfy incorrect tests regardless of code correctness. This empirically justifies our plateau heuristic as a necessary early-stopping mechanism designed to prevent the generator from wasting computation by chasing these unattainable signals.

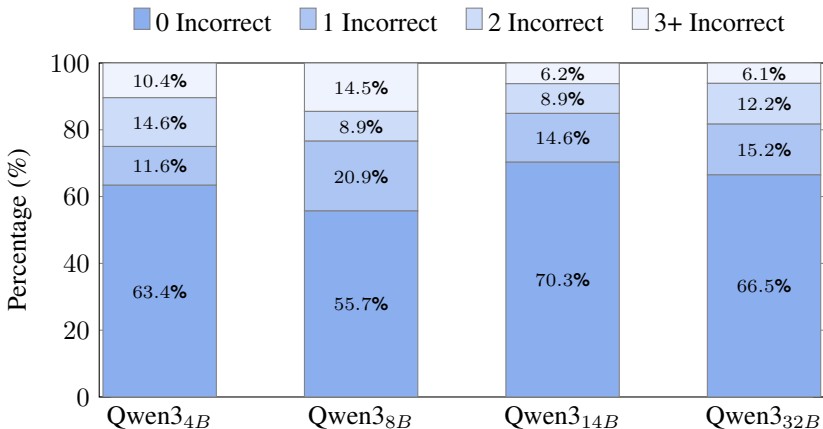

Figure 6: **Distribution of incorrect test cases (false positives) per problem on HumanEval.** The percentage of problems where the generated test cases contains 0, 1, 2, or 3+ incorrect test cases.

## 6 CONCLUSION

In this work, we introduced Planning-after-Trial (PaT), an adaptive policy that addresses the critical challenge of high inference costs in LLM-based code generation. By inverting the conventional Planning-before-Trial (PbT) policy, PaT avoids unnecessary planning overhead on simple problems and strategically allocates computational resources only when a direct attempt fails. Our theoretical analysis established the conditions for PaT's efficiency, particularly within a heterogeneous configuration that pairs a cost-efficient generator with a powerful planner. Comprehensive experiments confirmed that PaT outperforms existing state-of-the-art baselines in homogeneous settings across diverse models and benchmarks. We then empirically validated our theory, showing that a heterogeneous model configuration can further enhance efficiency. By enabling principled test-time computation scaling, PaT provides a practical and effective framework for building more scalable and cost-efficient code generation systems.

**Limitations**   Relying on self-generated test cases inherently introduces noise and potential inefficiency arising from unnecessary planning for correctly solved problems. While integrating emerging high-fidelity generation methods (Wang et al., 2025b; Prasad et al., 2025) offers synergistic potential, achieving a perfectly noise-free oracle remains unlikely; thus, robust mechanisms like our plateau heuristic remain essential for practical deployment. Regarding computational resources, our heterogeneous configuration entails a higher static memory footprint. However, this design aligns with efficient architectures like Mixture-of-Experts (MoE) (Jiang et al., 2024a), prioritizing the significant reduction of average dynamic inference cost and user-facing latency for the majority of tasks.

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

# A    THEORETICAL ANALYSIS

This section provides the detailed proof for Theorem 1, which was presented in Section 4. We then present and prove two additional theorems that extend our analysis: one that considers the asymptotic cost for problems of large complexity $k$, and another that derives the optimal cost for the generator sLM under scaling laws.

We restate the core assumptions from Section 4 that define our cost model.

**Assumption 1** (Generation cost). *A generation call by $M_G$ on a problem of complexity $k$ incurs*

$$Cost_{Generation} = \begin{cases} k\,c_{M_G}, & k \le p_{M_G} \quad (\text{success}), \\ p_{M_G}\,c_{M_G}, & k > p_{M_G} \quad (\text{failed attempt}). \end{cases} \tag{5}$$

**Assumption 2** (Planning cost). *A Planner-produced $n$-way plan maps a problem of size $k$ to $n$ independent subproblems of size $\frac{k}{n}$ and incurs a fixed per-subproblem overhead $D_{M_P}$, so that*

$$Cost_{Planning} = nD_{M_P}. \tag{6}$$

To formally connect a model's capability $p_M$ and its unit cost $c_M$, we adopt a standard assumption from the literature on neural scaling laws, which is empirically validated for Qwen3 model family in Figure 7.

**Assumption 3** (Scaling law). *We assume that a model's capability $p_M$ and its unit cost $c_M$ are related by a power-law scaling relation, consistent with prior work (Kaplan et al., 2020):*

$$p_M = \alpha\,c_M^\beta, \qquad \alpha > 0,\ 1 \ge \beta > 0. \tag{7}$$

*The constraint $0 < \beta \le 1$ reflect the principle of diminishing returns.*

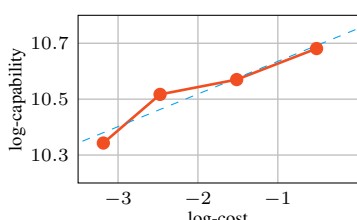

Figure 7: **Empirical validation of the scaling law (HumanEval+).** The linear fit in log-log space (capability vs. cost) validates our scaling law assumption in equation 7.

Also for convenience, we restate Theorem 1 before its proof.

**Theorem 1** (Existence of an efficient sLM capability). *Let $k \sim \text{Unif}(0, p_L)$. Under Assumptions 1, 2, and 3, if the total planning overhead satisfies*

$$nD_L \; < \; \left(\tfrac{1}{2} - \tfrac{1}{n^2}\right) p_L c_L, \tag{8}$$

*then there exists $p_s \in (0, p_L)$ for which PaT with a Heterogeneous Model Configuration has strictly lower expected cost than an LLM-only policy.*

*Proof.* First, for the homogeneous LLM-only policy, the cost for any problem $k \le p_L$ is $kc_L$. The expected cost is therefore:

$$\mathbb{E}[\text{Cost}_{\text{Homogeneous}}] = \frac{1}{p_L}\int_0^{p_L} kc_L\,dk = \frac{1}{p_L}\frac{p_L^2 c_L}{2} = \frac{p_L c_L}{2} \tag{9}$$

Next, for the Heterogeneous policy, we consider two cases based on the capability of the small model, $p_s$. If $p_s \ge k$, the generator $s$ succeeds, incurring a cost of $kc_s$. Else, *i.e.*, $p_s < k$, the generator $s$ fails (cost $p_s c_s$), the planner $L$ is invoked (cost $nD_L$), and then the generator $s$ solves the $n$ subproblems of size $\frac{k}{n}$. Consider $p_s \ge \frac{p_L}{n}$, we can calculate the cost in this case $p_s c_s + nD_L + kc_s$. The expected cost is the sum of the integrals over these two ranges, divided by $p_L$:

$$\mathbb{E}[\text{Cost}_{\text{Heterogeneous}}] = \frac{1}{p_L}\left(\int_0^{p_s} kc_s\,dk + \int_{p_s}^{p_L} p_s c_s + nD_L + kc_s\,dk\right) \tag{10}$$

$$= \frac{1}{p_L}\left(\frac{p_s^2 c_s}{2} + p_s c_s(p_L - p_s) + nD_L(p_L - p_s) + \frac{(p_L^2 - p_s^2)c_s}{2}\right) \tag{11}$$

$$= \frac{p_L c_s}{2} + \frac{p_L - p_s}{p_L}\left(p_s c_s + nD_L\right) \tag{12}$$

To analyze a concrete scenario, let's assume we can choose an sLM such that its capability is a fraction of the LLM's, *i.e.*, $p_s = \frac{p_L}{n}$. Using the scaling law from Assumption 3 ($p_M = \alpha c_M^\beta$), we can relate the costs: $c_s = \left(\frac{p_s}{\alpha}\right)^{1/\beta} = \left(\frac{p_L}{n\alpha}\right)^{1/\beta} = n^{-1/\beta} c_L$.

Substituting these into the heterogeneous cost equation equation 12:

$$\mathbb{E}[\text{Cost}_{\text{Heterogeneous}}] = \frac{p_L c_s}{2} + \frac{p_L - p_s}{p_L}\left(p_s c_s + n D_L\right) \tag{13}$$

$$= \frac{p_L n^{-1/\beta} c_L}{2} + \frac{n-1}{n}\left(\frac{p_L}{n} n^{-1/\beta} c_L + n D_L\right) \tag{14}$$

$$= \left(\frac{1}{2} + \frac{n-1}{n^2}\right) n^{-1/\beta} p_L c_L + (n-1) D_L \tag{15}$$

For the heterogeneous policy to be strictly more efficient, the difference must be positive:

$$0 > \mathbb{E}[\text{Cost}_{\text{Heterogeneous}} - \text{Cost}_{\text{Homogeneous}}] \tag{16}$$

$$= (n-1) D_L + \left(\frac{1}{2} + \frac{n-1}{n^2}\right) n^{-\frac{1}{\beta}} p_L c_L - \frac{p_L c_L}{2} \tag{17}$$

Rearranging to solve for the planning overhead $nD$ gives the condition stated in the theorem:

$$n D_L < \frac{n}{n-1}\left(\frac{p_L c_L}{2} - \left(\frac{1}{2} + \frac{n-1}{n^2}\right) n^{-\frac{1}{\beta}} p_L c_L\right) = \left(\frac{n - n^{1-\frac{1}{\beta}}}{2(n-1)} - n^{-1-\frac{1}{\beta}}\right) p_L c_L. \tag{18}$$

As a specific intuitive case, if we consider linear scaling where $\beta = 1$, the condition simplifies to:

$$n D_L < \left(\frac{1}{2} - \frac{1}{n^2}\right) p_L c_L \tag{19}$$

This shows that as long as the total planning cost is less than the savings achieved by the heterogeneous configuration, a more efficient sLM exists. □

**Theorem 2** (Asymptotic efficiency of the heterogeneous configuration). *For any sufficiently complex task $k$, the Heterogeneous strategy is asymptotically more cost-efficient than the Homogeneous strategy, provided that the cost of decomposition satisfies the following condition:*

$$D_L < \frac{c_L - c_s}{\frac{1}{p_s} - \frac{1}{p_L}}. \tag{20}$$

*Proof.* We prove the theorem by analyzing the asymptotic cost of each strategy for a large problem of complexity $k$. Let $h_M = \lceil k/p_M \rceil$ be the number of recursive division levels for a model $M$. The total number of division operations is $1 + n + n^2 + ... + n^{h_M-1} = \frac{n^{h_M}-1}{n-1}$. For a large $k$, we can approximate it as $\frac{n^{h_M}-1}{n-1} \approx \frac{k}{p_M(n-1)}$.

The total cost for a model $M$ is the sum of three components: the cumulative cost of failures at each division step $\frac{k}{p_M(n-1)} p_M c_M$, the cumulative cost of decomposition $\frac{k}{p_M(n-1)} n D_L$, and the final cost of conquering the sub-problems $k c_M$. Summing these components gives the total asymptotic cost:

$$\frac{nk}{n-1}\left(c_M + \frac{D_L}{p_M}\right). \tag{21}$$

For the heterogeneous setting to be more cost-efficient than the homogeneous one, it requires:

$$\frac{nk}{n-1}\left(c_s + \frac{D_L}{p_s}\right) < \frac{nk}{n-1}\left(c_L + \frac{D_L}{p_L}\right). \tag{22}$$

The $\frac{nk}{n-1}$ term cancels. Rearranging the remaining terms to solve for $D_L$ yields the condition stated in the theorem:

$$D_L < \frac{c_L - c_s}{\frac{1}{p_s} - \frac{1}{p_L}}. \tag{23}$$

□

This theorem provides a theoretical basis for the efficiency of heterogeneous model configuration. It indicates that the heterogeneous configuration becomes more cost-efficient when the cost of decomposition ($D_L$) is less than the savings generated by executing the sub-problems with a more cost-effective model ($s$). This provides a formal rationale for using the heterogeneous configuration and shows that the PaT policy is a structured approach to allocating computational resources.

**Theorem 3** (Optimal Generator cost under scaling laws). *The cost of the optimal small model, $c_s^*$, that minimizes the asymptotic cost of the heterogeneous configuration is given by the following closed-form solution:*

$$c_s^* = \min \left\{ \left( \frac{\beta \cdot D_L}{\alpha} \right)^{\frac{1}{\beta+1}}, c_{LLM} \right\} \tag{24}$$

*Proof.* The asymptotic cost of the heterogeneous configuration is proportional to the coefficient $c_s + \frac{D_L}{p_s}$. Substituting the scaling law gives $c_s + \frac{D_L}{\alpha c_s^\beta}$. To find the minimum, we take the derivative with respect to $c_s$ and set it to zero:

$$1 - \frac{\beta D_L}{\alpha} c_s^{-\beta-1} = 0. \tag{25}$$

Solving for $c_s$ yields the unconstrained optimum. The final solution is capped at $c_L$ to respect the problem's practical constraints. $\square$

This theorem provides a practical, closed-form solution for the cost of the optimal generator model in the asymptotic regime. The result serves as a powerful heuristic to approximately estimate the most cost-effective smaller model to use in real-world heterogeneous configurations.

**Empirical Validation.** To validate the practical utility of Theorem 3, we applied the formula to our HumanEval experimental data using Qwen3 models. We modeled model capability $p$ against normalized input token costs $c \in \{0.11, 0.18, 0.35, 0.70\}$ using the scaling law $p = \alpha c^\beta$. Fitting this to our observed data yielded $\alpha \approx 1722.2$ and $\beta \approx 0.12$, with an average planning cost $D_L \approx 1270.73$ derived from the heterogeneous (4B+32B) experiments. Substituting these parameters into Theorem 3 yields a theoretical optimal cost $c_s^* \approx 0.114$. This value is remarkably close to the actual cost of the 4B model ($c = 0.11$). While our main experiments (Figure 5) suggest the 8B model offers a strong performance-cost balance, the 4B model is indeed the strictly cost-optimal generator as predicted. This alignment confirms that despite simplified assumptions, our theoretical model successfully captures the underlying cost dynamics and serves as a practical guideline for selecting the initial sLM size for hyperparameter search.

## B   IMPLEMENTATION DETAILS

**Generation.**   The trial phase of our PaT policy incorporates a Best-of-N strategy to maximize the chance of a direct success. For each problem specification, the generator model $M_G$ produces 5 candidate solutions, using a temperature of 0.8 to encourage diverse outputs. Each of these 5 candidates is then verified against the test set. If any of the candidates passes all test cases, the process terminates successfully, and that solution is returned. Only if all 5 candidates fail the verification step does the policy escalate to the planning phase. The generation is retried up to 3 times if it fails to produce a parsable output.

**Planning.**   If the trial fails, the planning phase is triggered. The planner model $M_P$ is prompted to decompose the problem specification $x$ by rewriting the main function to call new unimplemented helper functions. A single decomposition plan is generated with a temperature of 0.2. From the resulting Python code block, we parse the new function signatures and docstrings, which become the specifications for the subproblems. If the output is not a valid code block, this planning step is retried up to 3 times, and the maximum recursion depth is limited to 3 to prevent overly complex solutions. Once all subproblems are recursively solved, their solutions are composed into a final program and verified against the original test suite for $x$. If this composite solution still fails, PaT initiates a re-planning loop. The planner is invoked again on the original problem $x$, but this time it is provided with the set of previously successful helper functions as additional context, enabling a more informed and cost-efficient re-planning attempt.

**Test case generation & verification.**   Our verification process requires a test suite $\mathcal{T}(x)$ for each problem $x$, which we construct in two stages. First, we process any example test cases provided directly in the problem description. Then, to ensure a comprehensive and robust evaluation, we augment the initial suite by prompting an LLM to generate additional test cases based on the problem description, a technique inspired by CodeT. This test generation process is performed consistently with a temperature of 0.2 and is retried up to 3 times.

# C BENCHMARK DETAILS

**HumanEval** (Chen et al., 2021) is a foundational dataset for evaluating the functional correctness of generated code. It consists of 164 hand-written programming problems, each including a function signature, a detailed docstring, and a set of hidden unit tests for evaluation. This benchmark is the standard for measuring the code generation capabilities of a model.

**MBPP** (Chen et al., 2022b) is a larger, crowd-sourced dataset. A critical challenge with the original MBPP dataset is that the prompts contain the ground-truth test cases, which can cause label leakage, particularly for baselines that perform selection or refinement. To ensure a fair comparison, we follow the setup of Shinn et al. (2023). For our experiments, we use a representative subset of 175 problems sampled from the full dataset.

**EvalPlus** (Liu et al., 2023) ensures a rigorous and reliable evaluation by augmenting the standard test suites of both HumanEval and MBPP. HumanEval and MBPP sometimes pass solutions that are functionally correct on the provided tests but fail on more subtle edge cases. EvalPlus mitigates this risk by automatically generating a much larger and more comprehensive set of unit tests, providing a more robust measure of a candidate solution's true correctness.

**xCodeEval** (Khan et al., 2023) is more challenging, competition-level problems, which is sourced from the CodeForces platform. For our experiments, we sample a subset of 500 problems, after first filtering out any instances with incomplete or invalid test cases. A key feature of xCodeEval is its difficulty labels, which allow us to partition problems into four categories (Easy, Mid, Hard, and Expert) for a more fine-grained analysis of policy behavior.

# D    LLMs Details

We selected a range of state-of-the-art open-source language models to evaluate our PaT policy across different architectures and scales. All experiments were conducted using NVIDIA A 6000 GPUs, with models served via the vLLM framework at float16 precision. Due to its memory size, the Qwen3$_{32B}$ model was run using 2-way tensor parallelism across two A6000 GPUs.

**Qwen3**   (Yang et al., 2025) As our primary model family, we use the Qwen 3 series in four sizes: 4B, 8B, 14B, and 32B. For all experiments, we used the base, pre-trained versions of these models.

**Llama-3.1**   (Dubey et al., 2024) To test the cross-family generalization of our approach, we use Llama-3.1-8B-Instruct. This is the instruction-tuned version of the Llama-3.1 8B model, optimized for following user commands and prompts.

**DeepSeek-Coder-V2-Lite**   (Zhu et al., 2024) To evaluate on a model specifically fine-tuned for code generation, we use DeepSeek-Coder-V2-Lite-Instruct. This is the instruction-tuned "Lite" version of the DeepSeek-Coder-V2 family, with approximately 16B parameters.

Table 4: **Baseline details.** Hyperparameter details for baselines and PaT.

| Method | Standard | Best-of-N | CodeT | FunCoder | PaT |
|---|---|---|---|---|---|
| **Generation** | | | | | |
| Samples (N) | 1 | 5 | 11 | 1 + 10 | 5 |
| Temperature | 0.3 | 0.8 | 0.8 | 0.8 | 0.8 |
| Retries | 3 | 3 | 3 | 3 | 3 |
| **Planning** | | | | | |
| Samples | - | - | - | 1 | 1 |
| Temperature | - | - | - | 0.2 | 0.2 |
| Retries | - | - | - | 3 | 3 |
| **Test case generation & verification** | | | | | |
| Benchmark-provided | X | O | X | O | O |
| Generation | X | X | O | O | O |
| Temperature | - | - | 0.2 | 0.2 | 0.2 |
| Retries | - | - | 3 | 3 | 3 |

# E    BASELINE DETAILS

**Standard**    (Brown et al., 2020) represents the most direct approach to code generation, establishing the base capability of a model. It performs a single, one-time generation attempt to produce the entire program from the problem specification. For our code generation tasks, we follow a few-shot prompting protocol, providing the model with a small number of in-context examples to guide its output. We generate a single candidate solution with a low temperature of $0.3$ to produce the most probable and deterministic result. This baseline serves to measure the model's raw performance without any complex strategies like sampling, refinement, or decomposition.

**Best-of-N**    (Chen et al., 2021) aims to improve performance by exploring a diverse set of potential solutions through sampling. For each problem, it generates $N$ candidate programs using a high temperature to encourage variety. To ensure a fair comparison with the trial phase of our PaT policy, we use the same sampling parameters: we set $N = 5$ and use a temperature of $0.8$. Each of the 5 candidates is then executed against the provided test suite, and the one that passes the most tests is selected as the final solution. This method increases the probability of finding a correct solution at the cost of generating and evaluating multiple candidates. Since MBPP does not have a test suite, we deterministically submit the first of the 5 generated candidates.

**CodeT**    (Chen et al., 2022a) is utilized from the implementation provided by the authors of Fun-Coder to ensure a faithful reproduction of their setup. The process begins by sampling a pool of candidate solutions. Following the hyperparameter settings of FunCoder, we used a larger sample size of $N = 11$ with a temperature of $0.8$, which diverges from PaT and Best-of-N's $N = 5$. This is because CodeT's consensus-based ranking mechanism is highly dependent on a large and diverse candidate pool to function effectively; using a smaller sample size would artificially weaken this baseline.

**FunCoder**    (Chen et al., 2024) serves as our primary baseline for the Planning-before-Trial (PbT) policy, and we use the official implementation to ensure a fair and accurate comparison. The method operates as a rigid two-stage pipeline. First, a planner model decomposes the problem into a complete plan of helper functions using a temperature of $0.8$. Only after this static plan is finalized does it proceed to the second stage, where it solves each subproblem using a consensus-based mechanism similar to CodeT, with $N = 11$ and a temperature of $0.8$. This rigid, plan-first approach, where the plan is fixed regardless of generation outcomes, provides a direct contrast to our adaptive PaT policy.

# F PROMPT DETAILS

Our prompting strategy largely follows the one provided in the FunCoder (Chen et al., 2024) implementation to ensure a fair comparison. We use the same few-shot examples and sampling methods for both the generation (conquer) and planning (divide) phases.

```
...
- If a single function is too hard to solve, you can decompose it into
    multiple smaller functions.
...
```
(a) Decomposition prompt for FunCoder (Chen et al., 2024)

```
...
- The previous attempt to direct implement the target function is failed
    , indicating its overall logic might be too complex to implement
    directly.
- Therefore, you must decompose it into multiple smaller, manageable
    helper functions.
...
```
(b) Planning prompt for PaT

Figure 8: **Comparison of Planning Prompts.** (a) An excerpt from a decomposition prompt using the pre-divided method, which instructs the model to decompose a function if it is too complex. (b) An excerpt from the PaT planning prompt, which provides more explicit instructions based on a failed execution, enabling a clearer and more direct decomposition strategy.

The single critical modification is in the decomposition prompt. The FunCoder prompt uses a conditional instruction, asking the planner to decompose a problem if it seems complex. In contrast, since our PaT policy only invokes the planner after a direct attempt has already failed, the problem's difficulty has been established. Therefore, our planner prompt is an unconditional command to decompose the problem. Additionally, for all experiments with the Qwen3 model family, we prepended the /no_think command to all prompts.

## F.1 PROMPT FOR GENERATION

```
You are a programming copilot, you can solve a problem by writing Python
    functions. Your task is to:

  - For every turn, you need to write a Python function that returns the
    answer, based on current code (not code in chat history) and problem
    description.
  - Do not modify function name, arg names, docstring in given functions.
  - Consider reusing existing functions that are already implemented.
  - You can import libraries to better solve the problem.

<User>:

Current Code:

```python
def prime_factor(x: int) -> list:
    """get a list of prime factors of number $x$"""
    ret = []
    i = 1
    while i * i <= x:
        i += 1
        if x % i == 0 and is_prime(i):
            ret.append(i)
    return ret

def is_prime(x: int) -> bool:
    """determine $x$ is a prime number or not"""
```

```
    if x < 2:
        return False
    for i in range(2, int(x**0.5) + 1):
        if x % i == 0:
            return False
    return True

def get_common(a: list, b: list) -> list:
    """get common element in two list $a$ and $b$"""
    ret = []
    for item in a:
        if item in b:
            ret.append(item)
    return ret

def sum_common_factors(a: int, b: int) -> int:
    """Return the sum of all common prime factors of $a$ and $b$"""

    raise NotImplementedError()
```

Let's think step by step and implement the following method `
    sum_common_factors` using existing functions to solve:
"Return the sum of all common prime factors of $a$ and $b$"

<Assistant>:

First, I need to get the prime factors of $a$ and $b$.
Second, I can use `for` loop to find common element in two factors list.
Finally, sum the common factor list and return the answer.
Here is the `sum_common_factors` function:

```python
def sum_common_factors(a: int, b: int) -> int:
    """Compute the sum of all common prime factors of $a$ and $b$"""
    factors_a = prime_factor(a)
    factors_b = prime_factor(b)
    common_factors = get_common(factors_a, factors_b)
    return sum(common_factors)
```

<User>:

Current Code:

```python
{prev_code}
```

Let's think step by step and implement the following method `{
    cur_func_name}` using existing functions to solve:
"{cur_func_doc}"

## F.2 PROMPT FOR PLANNING

You are a programming copilot, you can solve a problem by writing Python
    functions. Your task is to:
  - The previous attempt to direct implement the target function is
    failed, indicating its overall logic might be too complex to
    implement directly.
  - For every turn, you need to write a Python function that returns the
    answer based on Current Code (not code in chat history).
  - Do not modify function name, arg names, docstring in given functions.
  - You can import libraries to better solve the problem.
  - You can leave new function unimplemented for now, but write the
    function at the end of the code and comment what the function does.

```
  - Therefore, you must decompose it into multiple smaller, manageable
    helper functions.

<User>:

Current Code:
'''python
def sum_common_factors(a: int, b: int) -> int:
    """Compute the sum of all common prime factors of $a$ and $b$"""
    raise NotImplementedError()
'''

Let's think step by step and complete the following Python function `
    sum_common_factors` that solves:
"Compute the sum of all common prime factors of $a$ and $b$"

<Assistant>:

First, I need to get the prime factors of $a$ and $b$.
Second, I can use 'for' loop to find common element in two factors list.
Finally, sum the common factor list and return the answer.
Here is the 'sum_common_factors' function:

'''python
def sum_common_factors(a: int, b: int) -> int:
    """Compute the sum of all common prime factors of $a$ and $b$"""
    factors_a = prime_factor(a)
    factors_b = prime_factor(b)
    common_factors = get_common(factors_a, factors_b)
    return sum(common_factors)

def prime_factor(x: int) -> list:
    """get a list of prime factors of number $x$"""
    raise NotImplementedError()

def get_common(a: list, b: list) -> list:
    """get common element in two list $a$ and $b$"""
    raise NotImplementedError()
'''

<User>:

Current Code:
'''python
def sum_common_factors(a: int, b: int) -> int:
    """Compute the sum of all common prime factors of $a$ and $b$"""
    factors_a = prime_factor(a)
    factors_b = prime_factor(b)
    common_factors = get_common(factors_a, factors_b)
    return sum(common_factors)

def get_common(a: list, b: list) -> list:
    """get common element in two list $a$ and $b$"""
    raise NotImplementedError()
'''

Let's think step by step and complete the following Python function `
    get_common` that solves:
"get common element in two list $a$ and $b$"

<Assistant>:

Here is the 'get_common' function:

'''python
```

```python
def get_common(a: list, b: list) -> list:
    """get common element in two list $a$ and $b$"""
    ret = []
    for item in a:
        if item in b:
            ret.append(item)
    return ret
```

<User>:

Current Code:
```python
{prev_code}
```

Let's think step by step and complete the following Python function `{cur_func_name}` that solves:
"{cur_func_doc}"

# G  ADDITIONAL RESULTS

Table 5: **Full results on heterogeneous setting.** We report the average cost relative to Qwen3$_{32B}$ is normalized to 1.00.

| Generator | Planner | HumanEval | | MBPP | | Avg. | $\Delta$ Avg. | cost |
|---|---|---|---|---|---|---|---|---|
| | | base | plus | base | plus | | | |
| Qwen3$_{32B}$ | Qwen3$_{32B}$ | 93.90 | 84.15 | 88.57 | 86.86 | 88.37 | - | 1.00 |
| Qwen3$_{14B}$ | Qwen3$_{14B}$ | 91.46 | 83.54 | 85.14 | 82.29 | 86.18 | - 2.19 | 0.47 |
| | Qwen3$_{32B}$ | 93.29 | 85.98 | 86.29 | 84.57 | 87.53 | - 0.84 | 0.49 |
| Qwen3$_{8B}$ | Qwen3$_{8B}$ | 90.85 | 82.32 | 85.14 | 84.00 | 85.58 | - 2.79 | 0.25 |
| | Qwen3$_{32B}$ | 93.90 | 85.37 | 86.29 | 84.00 | 87.39 | - 0.98 | 0.31 |
| Qwen3$_{4B}$ | Qwen3$_{4B}$ | 89.63 | 82.32 | 82.29 | 78.29 | 83.13 | - 5.24 | 0.14 |
| | Qwen3$_{4B}$ | 92.07 | 84.76 | 82.29 | 80.00 | 84.78 | - 3.40 | 0.18 |

**Full version of Table 5a.**  Table 5 presents the full results for our heterogeneous setting experiments across all four foundational benchmarks (HumanEval, HumanEval+, MBPP, and MBPP+). In this configuration, we pair a series of smaller generator models with a powerful, fixed planner (Qwen3$_{32B}$). The table provides the detailed Pass@1 scores and relative costs for each combination, which were aggregated and visualized in Figure 5b in the main body of the paper.

## H    THE USE OF LARGE LANGUAGE MODELS (LLMs)

In the preparation of this manuscript, we utilized an LLM as a writing assistant to aid and polish the text. The LLM's role included refining the phrasing and grammar of author-written drafts, suggesting alternative sentence structures to improve clarity, and ensuring a consistent academic tone throughout the paper. All technical contributions, experimental designs, and final claims were conceived by the human authors. The authors reviewed, edited, and take full responsibility for all content presented in this paper.

