# OpenReview forum: "PaT: Planning-after-Trial for Efficient Code Generation"
_ICLR.cc/2026/Conference — ICLR 2026 Conference Withdrawn Submission_

### Official Review · Reviewer_NVMM · 2025-10-26

**Soundness:** 3
**Presentation:** 3
**Contribution:** 3
**Rating:** 2
**Confidence:** 5

**Summary:**

This paper proposes Planning-after-Trial (PaT) policy for code generation, where code generation with a cost-efficient generator (the "Trial") is followed after a powerful planner (the "Planning") if the initial attempt fails, which  reduces unnecessary planning overhead.

**Strengths:**

* Efficiency: By avoiding planning on a simple task, efficiency can be achieved
* Adaptivity: Policy adapts to problem difficulty.

**Weaknesses:**

* Dependency: The claimed efficiency improvement arises from skipping explicit planning for easier tasks. However, this benefit relies heavily on the quality initial generation. Sensitivity analysis would be useful.
* Negative generation: Analysis on initial failures potentially negating the cost advantage would be necessary. If early-stage errors are frequent, can total cost exceed that of a consistent planning-based approach.
* Cost: Overheads  of deploying two models need to  be further justified. More discussion on increased resource overhead (e.g., memory, latency) would be useful

**Questions:**

* Regarding dependency, can the policy adapt if the initial generation is systematically poor?
* Regarding negative generation, an total cost exceed that of a consistent planning-based approach.
* Are increased costs acceptable in real-world deployment?
* Could a single multitask model emulating both roles be an alternative?

---

> ### Author Response · Authors · 2025-11-20
>
> ## Weakness 1
> Dependency: The claimed efficiency improvement arises from skipping explicit planning for easier tasks. However, this benefit relies heavily on the quality initial generation. Sensitivity analysis would be useful.
>
> ## Question 1
> Regarding dependency, can the policy adapt if the initial generation is systematically poor?
>
> ## Weakness 2
> Negative generation: Analysis on initial failures potentially negating the cost advantage would be necessary. If early-stage errors are frequent, can total cost exceed that of a consistent planning-based approach.
>
> ## Question 2
> Regarding negative generation, an total cost exceed that of a consistent planning-based approach.
>
> ---
>
> ## Answer
> We agree that our PaT policy’s cost efficiency is critically dependent on the sLM's quality, acknowledging that frequent sLM failure (Negative Generation) risks having the total cost exceed that of a consistent planning-based approach. This inherent trade-off is precisely why our theoretical analysis in Section 4.1 is a central contribution. We first theoretically provided the existence of a cost-efficient sLM (Theorem 1) by establishing the necessary condition that the cost of problem decomposition must be below a certain value. This theoretical framework, fully detailed in Theorem 3, then allows us to rigorously derive the necessary cost-capability threshold required for the heterogeneous configuration to achieve superior efficiency, providing the concrete mathematical procedure for determining the optimal sLM size. We have added the empirical calculation of this Theorem 3 based on our experimental results to Appendix A. This analysis confirms that the policy's adaptability is rooted in a mathematically justifiable design, as we can select the sLM required to meet this threshold, making the cost-efficiency a design constraint rather than a risk.
>
> ---
>
> ## Weakness 3
> Cost: Overheads of deploying two models need to be further justified. More discussion on increased resource overhead (e.g., memory, latency) would be useful
>
> ## Question 3
> Are increased costs acceptable in real-world deployment?
>
> ---
>
> ## Answer
> The overhead associated with deploying our two-model system (LLM + sLM) is justified by the significant long-term operational value gained from the conditional resource allocation. While the total memory footprint is indeed higher than that of a single LLM approach, we argue that this trade-off is acceptable and highly valuable for real-world deployment. The current trend in large model architectures, such as Mixture-of-Experts (MoE), highlights that the industry is shifting focus from merely minimizing total deployed parameters to optimizing the average number of active parameters and the total inference cost/latency. Our PaT policy operates on a similar principle: the high static memory overhead is justified by the significant dynamic cost savings achieved by avoiding the LLM’s expensive computation for a substantial portion of the workload. Furthermore, from a user experience perspective, our policy offers a distinct advantage as utilizing the sLM for easier tasks provides significantly lower latency for common queries, enhancing user satisfaction. Therefore, the increased static resource overhead is justified by the reduction in average operational cost and improved user-facing latency on simple tasks.
>
> ---
>
> ## Question 4
> Could a single multitask model emulating both roles be an alternative?
>
> ---
>
> ## Answer
> Yes, this is explicitly addressed in our Section 5.1, 'Homogeneous Setting' experiments. We showed that the PaT policy's core strength is its Adaptive logic, which demonstrated superior performance even when a single high-performance model performed both the Generation and Planning roles. This confirms the validity of the policy structure itself, independent of the cost-saving benefits of the sLM.

---

> > ### Comment · Reviewer_NVMM · 2025-11-26
> >
> > I rased score as responses addressed concerns. Claims in appendix/rebuttal can be moved to main text in the revision.

---

> > > ### Author Response · Authors · 2025-11-27
> > >
> > > We sincerely thank the reviewer for the constructive feedback. As suggested, we have moved the quantitative analysis of generated test cases to the main text (Section 5.3) in the revised manuscript. We have also added a detailed discussion regarding verification robustness and increased resource overhead to the Limitations paragraph in the Conclusion (Section 6). We remain fully available for any further discussion or questions you may have.

---

### Official Review · Reviewer_nMUT · 2025-10-28

**Soundness:** 2
**Presentation:** 2
**Contribution:** 1
**Rating:** 2
**Confidence:** 5

**Summary:**

This paper proposes PaT (Planning-after-Trial) to improve the efficiency of LLM code generation. Unlike the Planning-before-Trial (PbT), PaT first attempts a direct solution and only invokes the Planner upon verification failure. The authors further propose that PaT combined with a "heterogeneous configuration" (using a small model for generation and a large model for planning), can achieve performance close to that of the large model at a fraction of the cost.

**Strengths:**

The paper is clearly written and easy to follow. Improving the inference efficiency of LLM code generation is of significant practical importance and broad interest.

**Weaknesses:**

1. The core idea (PaT) is an incremental improvement over the PbT policy. More critically, the method is highly similar to AdaCoder [1], (arXiv: 2504.04220), which also proposes an adaptive planning strategy. Given that this paper cites later work from September 2025 (Paglieri et al.), [1] should not be considered concurrent work. The authors must clarify their contribution's novelty relative to [1].

2. Method 3.2 relies on a model-generated unit-test set for verification, which is a highly problematic step. As noted in [5, 6], model-generated test oracles are prone to errors and can lead to "false positives," thus misleading subsequent generation. The authors' "plateau hypothesis" (i.e., assuming remaining failures are due to flawed tests when $p^{(t)} \le p^{(t-1)}$) is an unsubstantiated and unrigorous heuristic.

3. The theoretical analysis in Section 4.1 and Appendix A reads more like "mathematical decoration" than a source of practical insight. The theory is built on highly simplified, unmeasurable abstractions (e.g., representing "complexity" as a scalar $k$ and "capability" as $p$). Theorem 3 claims a "practical, closed-form solution" to guide the selection of the optimal generator (sLM) is entirely unsubstantiated empirically. The authors never attempt to use their own data (e.g., Fig 6) to estimate the formula's parameters and, in turn, predict their own experimental results (i.e., why Qwen3-8B was the optimal choice). This makes the theoretical section appear superfluous and unconvincing.

4. The DIVIDE and COMPOSE procedure in Method 3.1 is nearly identical to the hierarchical decomposition mechanisms in FunCoder and CodeChain [7]. The authors fail to clarify its novelty. And paper lacks necessary citations and discussion of related work[2,3,4].

[1] AdaCoder: An Adaptive Planning and Multi-Agent Framework for Function-Level Code Generation https://arxiv.org/abs/2504.04220

[2] Planning In Natural Language Improves LLM Search For Code Generation https://arxiv.org/abs/2409.03733

[3] INTERVENOR: Prompting the Coding Ability of Large Language Models with the Interactive Chain of Repair https://arxiv.org/abs/2311.09868

[4] PairCoder: A Pair Programming Framework for Code Generation via Multi-Plan Exploration and Feedback-Driven Refinement https://arxiv.org/abs/2409.05001

[5] TESTEVAL: Benchmarking Large Language Models for Test Case Generation https://arxiv.org/abs/2406.04531

[6] Learning to Generate Unit Tests for Automated Debugging https://arxiv.org/abs/2502.01619

[7] CodeChain: Towards Modular Code Generation Through Chain of Self-revisions with Representative Sub-modules https://arxiv.org/abs/2310.08992

**Questions:**

1. Did you measure the "false positive/negative" rate of the model-generated test cases? How does your "plateau heuristic" perform in the face of these erroneous test signals? For example, can it distinguish between a failure caused by a code bug versus a failure caused by a test bug?

2. Can you empirically apply Theorem 3's formula using your own data (e.g., by fitting $$\alpha, \beta$$ from Fig 6 and estimating the average planning cost $D_L$)? Does the predicted theoretical optimal generator cost $c_s^*$ align with the optimal empirical result you observed in Section 5.2 (the $Qwen3_{8B}$ model)? If not, does this imply the theoretical model is too oversimplified to be useful for real-world deployment?

---

> ### Author Response · Authors · 2025-11-20
>
> ## Weakness 1
> The core idea (PaT) is an incremental improvement over the PbT policy. More critically, the method is highly similar to AdaCoder [1], (arXiv: 2504.04220), which also proposes an adaptive planning strategy. Given that this paper cites later work from September 2025 (Paglieri et al.), [1] should not be considered concurrent work. The authors must clarify their contribution's novelty relative to [1].
>
> ---
>
> ## Answer
> We sincerely thank the reviewer for identifying AdaCoder [1], which we had previously missed and have now added to our revised related work section. We agree our high-level "planning after trial" concept partially overlaps with AdaCoder. However, this adaptive strategy, attempting a simple, low-cost solution before escalating to an expensive, high-fidelity one, is a well-established pattern used in many fields, such as model cascading [8] and hierarchical RL [9]. PaT's novelty is established in the specific implementation details where it differs significantly from AdaCoder:
>
> 1. Planning Mechanism: The role of 'Planning' is different. AdaCoder's ‘Planning' focuses on code refinement, using failure signals to generate a patch. This is effective for simpler bugs but may struggle with tasks that are fundamentally complex relative to the given model's capability. PaT's 'Planning' is defined as hierarchical decomposition, breaking such a complex problem into simpler, solvable subproblems. This decomposition is well-suited for these harder tasks and serves as a key part of our framework, where both the how and when of adaptive planning are jointly determined. Also, we instantiate this with a heterogeneous model configuration as an effective implementation.
>
> 2. Test case: The verification methods highlight a crucial design difference. AdaCoder relies on pre-existing test oracles. This is problematic, as HumanEval's public tests are often too simple and miss edge cases, while MBPP provides none at all, forcing such methods to unrealistically use the hidden test case as an oracle. PaT is built for this oracle-scarce reality; we propose the self-generation of test cases (and heuristics like our 'plateau') as a necessary, practical solution to ensure robust, real-world deployment.
>
> [8] CascadeBERT: A unified framework for adaptive BERT prediction https://arxiv.org/abs/2012.14682
>
> [9] Breadth-First Exploration on Adaptive Grid for Reinforcement Learning https://proceedings.mlr.press/v235/yoon24d.html
>
> ---
>
>
> ## Weakness 2
> Method 3.2 relies on a model-generated unit-test set for verification, which is a highly problematic step. As noted in [5, 6], model-generated test oracles are prone to errors and can lead to "false positives," thus misleading subsequent generation. The authors' "plateau hypothesis" (i.e., assuming remaining failures are due to flawed tests when $p^{(t)} <= p^{(t-1)}$) is an unsubstantiated and unrigorous heuristic.
>
> ## Question 1
> Did you measure the "false positive/negative" rate of the model-generated test cases? How does your "plateau heuristic" perform in the face of these erroneous test signals? For example, can it distinguish between a failure caused by a code bug versus a failure caused by a test bug?
>
> ---
>
> ## Answer
> We thank the reviewer for this critical question, which addresses the core challenge of real-world verification. We fully acknowledge the concerns from [5, 6] and agree that model-generated test oracles are imperfect and prone to errors. To answer Question 1 directly, we did measure the "false positive" rate of our generated tests on the HumanEval benchmark and reported it in Appendix G. The analysis (detailed in Figure 8) confirms the verification signal's overall high quality. The results demonstrate that the majority of generated test suites (ranging from 55.7% to 70.3%) contain zero incorrect test cases, and when errors do occur, they are generally small in magnitude (i.e., concentrated in the 1 or 2 incorrect test categories).
>
> However, test case generation is essential because sole reliance on public tests is insufficient (lacking edge cases) or impossible (MBPP provides none). We view external advances on improving test fidelity [5, 6] as complementary and are excited about integrating them in future work, but we also believe that a perfectly noise-free oracle remains a very challenging, open problem.  Therefore, a practical framework must cope with this unavoidable noise. Our "plateau heuristic" is the pragmatic heuristic designed for this purpose: it acts as an early-stopping mechanism to prevent the system from wasting computation on what are likely spurious signals, a necessary feature for managing residual imperfections.

---

> > ### Author Response · Authors · 2025-11-20
> >
> > ## Weakness 3
> > The theoretical analysis in Section 4.1 and Appendix A reads more like "mathematical decoration" than a source of practical insight. The theory is built on highly simplified, unmeasurable abstractions (e.g., representing "complexity" as a scalar k and "capability" as p). Theorem 3 claims a "practical, closed-form solution" to guide the selection of the optimal generator (sLM) is entirely unsubstantiated empirically. The authors never attempt to use their own data (e.g., Fig 6) to estimate the formula's parameters and, in turn, predict their own experimental results (i.e., why Qwen3-8B was the optimal choice). This makes the theoretical section appear superfluous and unconvincing.
> >
> > ---
> >
> > ## Answer
> > We included the theoretical analysis (Sec 4.1) to formally address a key question that Reviewer NVMM also raised regarding negative generation cost and deployment overhead: is using the smallest possible model always the most cost-efficient? Intuitively, no, as a less capable model will trigger the expensive 'Planning' phase more often. We felt it was necessary to mathematically prove that a non-trivial, optimal sLM size exists.
> > We acknowledge the abstractions (k, p) are simplifications, but they are grounded in practice. Representing difficulty as a scalar (k) is common, mirroring the "Rating" systems on coding platforms. Based on the general principle that more capable models solve harder problems, we then define a model's capability (p) as the maximum difficulty k it can reliably solve. The fact that an LLM's performance follows a scaling law, relating model size and computational cost to performance, provides direct justification for representing this capability with a single scalar value p. This abstraction allowed us to formally prove the core finding that an optimal sLM exists.
> >
> > ---
> >
> > ## Question 2
> > Can you empirically apply Theorem 3's formula using your own data (e.g., by fitting \alpha, \beta) from Fig 6 and estimating the average planning cost  D_L)? Does the predicted theoretical optimal generator cost c_s^* align with the optimal empirical result you observed in Section 5.2 (the Qwen3_8B model)? If not, does this imply the theoretical model is too oversimplified to be useful for real-world deployment?
> >
> > ---
> >
> > ## Answer
> > We thank the reviewer for suggesting we validate this empirically. We have added this full analysis to Appendix A. To summarize our findings:For a more precise calculation, we went beyond the simple Pass@1 vs. Size plot in Figure 6. We calculated the parameters based on the HumanEval results, using the actual input token costs (0.11, 0.18, 0.35, 0.7) and the observed performance/token usage of the Standard models (4B-32B) to approximate p. The planning cost D_L was derived from the 4B+32B case.This calculation yielded alpha = 1722.2, beta = 0.12, and D_L = 1270.73. Plugging these into our formula, the predicted theoretical optimal cost c_s^* approximated as 0.114. This value is extremely close to the 4B model's actual cost (0.11), which suggests a model near the 4B scale is the most cost-efficient generator, aligning with our theorem's cost-centric goal.
> >
> > While the theory is not a perfect predictor (e.g., it assumes an asymptotic regime and does not consider the accuracy, considering cost only), we believe it serves as a practical guideline, offering a reasonable starting point for the hyperparameter search (i.e., selecting an sLM) when deploying PaT with new model families.

---

> > > ### Author Response · Authors · 2025-11-20
> > >
> > > ## Weakness 4
> > > The DIVIDE and COMPOSE procedure in Method 3.1 is nearly identical to the hierarchical decomposition mechanisms in FunCoder and CodeChain [7]. The authors fail to clarify its novelty. And paper lacks necessary citations and discussion of related work[2,3,4].
> > >
> > > ---
> > >
> > > ## Answer
> > > We thank the reviewer for this crucial feedback and apologize for the missing citations [2,3,4,7], which we have added to our revised Related Work section. We agree that the technical implementation of our DIVIDE/COMPOSE procedure is similar to the hierarchical mechanisms in FunCoder and CodeChain [7]. However, PaT's novelty lies in a different and higher level of adaptivity focused purely on cost-efficiency. In particular, we view PaT as a step toward more optimal test-time computation scaling across instances specifically tailored to code generation, a direction we believe is timely as computational costs for LLM inference continue to grow [10, 11]. CodeChain’s framework assumes a problem is complex and always enters its expensive, adaptive planning-and-revision loop, which is inherently wasteful for simple tasks. PaT’s core contribution is the ‘Trial’ phase (run by the sLM), which acts as a low-cost filter. This filter allows PaT to strategically bypass this entire expensive mechanism for the majority of easy problems, escalating to a decomposition (handled by the powerful LLM) only when an sLM’s ‘Trial’ failure empirically proves it is necessary. This makes PaT a distinct cost-optimization framework.
> > >
> > >
> > > [10] Scaling LLM Test-Time Compute Optimally Can be More Effective than Scaling Parameters for Reasoning https://arxiv.org/abs/2408.03314
> > >
> > > [11] Wider or Deeper? Scaling LLM Inference-Time Compute with Adaptive Branching Tree Search https://arxiv.org/abs/2503.04412

---

> > > > ### Comment · Reviewer_nMUT · 2025-11-26
> > > >
> > > > The revisions improve clarity but do not change my overall assessment. PaT’s main components --- adaptive planning, hierarchical decomposition, multi-model switching, and LLM-generated tests --- are already standard in prior work (e.g., AdaCoder, FunCoder, CodeChain), and the paper does not offer substantial new algorithmic insight beyond a repackaging of known ideas.
> > > >
> > > > On verification, even with the added analysis, a notable fraction of generated tests remain incorrect on HumanEval, despite Qwen3 models achieving >90% pass@1. Prior work (e.g., AgentCoder) has already highlighted this oracle-noise issue, and PaT’s plateau heuristic does not provide a principled way to handle misleading test signals. The core difficulty thus remains unaddressed.
> > > >
> > > > Theoretical results, though now instantiated with fitted parameters, remain narrow and largely a posteriori.
> > > >
> > > > Given these unresolved issues in novelty and soundness, I maintain my recommendation to reject.

---

> > > > > ### Author Response · Authors · 2025-11-26
> > > > >
> > > > > We genuinely appreciate your rigorous evaluation and the time taken to review our revisions. We would like to take this opportunity to clarify some possible misunderstandings regarding our contributions compared to prior work.
> > > > >
> > > > > ---
> > > > >
> > > > > ## Justification for Test Generation
> > > > >
> > > > > We agree that model-generated test cases can be noisy, but we maintain that they are necessary for a robust framework. Approaches that rely solely on provided test cases (e.g., AdaCoder) face fundamental limitations:
> > > > >
> > > > > * Unrealistic Availability: Some benchmarks, such as MBPP, do not provide public test cases in their prompts. To function in these settings, prior works often use hidden evaluation test cases for verification. We believe a practical system must be self-sufficient and not rely on peeking at hidden oracles.
> > > > >
> > > > > * Insufficient Coverage: Even when public test cases are available, they are often simple input-output examples that fail to cover critical edge cases. Relying solely on them creates a critical blind spot: the system may erroneously accept a solution that passes these simple checks but fails on complex logic. We argue that securing diverse test coverage via generation is worth the trade-off of managing potential noise.
> > > > >
> > > > > * Robustness against Error: In real-world scenarios, even provided specifications or test cases can be flawed. A system that blindly attempts to pass every test case risks overfitting to incorrect test cases.
> > > > >
> > > > > We view recent advances [5, 6] as complementary technologies that can be integrated into PaT to further enhance efficiency. However, we maintain that achieving a perfectly noise-free oracle is likely impossible; even with advanced algorithms, incorrect test cases will persist. In this context, our "plateau heuristic" acts as a necessary "early-stopping" mechanism, designed specifically to prevent the system from wasting resources by attempting to satisfy these few, incorrect test cases.
> > > > >
> > > > > ---
> > > > >
> > > > > ## Distinctiveness from Prior Works
> > > > >
> > > > > We view the synthesis of established strengths from prior work as a natural evolution in systems research. However, PaT is not a mere repackaging; it differentiates itself through critical implementation details.
> > > > >
> > > > > Regarding AdaCoder, the fundamental divergence lies in planning and verification. While AdaCoder relies on iterative refinement, PaT employs hierarchical decomposition to handle structurally complex problems that simple debugging cannot resolve. Furthermore, AdaCoder relies on provided oracle test cases, creating a critical vulnerability: public tests often lack edge cases, leading the system to erroneously accept incorrect solutions that pass simple checks but fail on complex logic. PaT mitigates this by generating comprehensive test suites. While generated tests may introduce noise (potential inefficiency), we prioritize this over the critical risk of accepting incorrect solutions (correctness failure). In our view, submitting a wrong answer due to blind spots is a far worse outcome than incurring slightly higher planning costs to ensure robustness.
> > > > >
> > > > > PaT differs significantly from FunCoder and CodeChain in terms of both efficiency and decomposition effectiveness. Unlike these methods, which rely on speculative pre-decomposition that risks creating suboptimal plans (e.g., insufficient granularity leading to failure), PaT is feedback-driven. It triggers decomposition only after a trial fails, ensuring that complex problems are broken down to a level that is actually solvable by the model. This allows PaT to succeed on hard tasks where the rigid, one-shot planning of prior works often fails. Consequently, PaT achieves superior performance than FunCoder with significantly optimized resource allocation, demonstrating substantial algorithmic gains.
> > > > >
> > > > > Furthermore, we propose a heterogeneous configuration (sLM + LLM) as a novel architectural choice to maximize cost-efficiency within this framework. Since the majority of tasks are resolved in the sLM-driven 'Trial' phase, this architecture significantly reduces both latency and inference cost for common queries, making adaptive code generation practical for real-world, low-latency deployment.
> > > > >
> > > > > ---
> > > > >
> > > > > ## Primary Role of Theoretical Analysis
> > > > >
> > > > > Theorem 1 constitutes our primary theoretical contribution, explicitly addressing the fundamental concern raised by Reviewer NVMM regarding the efficiency risks of small models. It formally establishes that a cost-efficient sLM size exists, providing the necessary theoretical grounding for our heterogeneous approach. While Theorem 3 is validated retrospectively, it serves as a practical guideline for selecting the optimal model size when training or deploying new models within a specific family.
> > > > >
> > > > > ---
> > > > >
> > > > > We hope these clarifications address your concerns, and we remain fully available for any further discussion or questions you may have.

---

### Official Review · Reviewer_KVew · 2025-10-29

**Soundness:** 3
**Presentation:** 2
**Contribution:** 2
**Rating:** 4
**Confidence:** 4

**Summary:**

This paper focuses on efficiency of code generation: existing approaches either skip planning (which may lead to poor performance on complex tasks) or adopt a "Plan-before-Trial" (PbT) paradigm that wastes computational resources on simple tasks by enforcing planning regardless of task difficulty.
To this end, the authors propose PaT (Planning-after-Trial), a two-stage framework that prioritizes efficiency without sacrificing performance.
In the first stage, PaT leverages lightweight, low-cost small models to directly generate code for a given task; it then validates the generated code using test cases. If the code passes validation, PaT terminates early to save costs.
If validation fails (indicating a complex task), PaT invokes a more capable but expensive large model to perform task decomposition (planning), breaking the complex task into manageable sub-tasks that are solved iteratively.
To further optimize efficiency, PaT adopts a strategy: small models handle code generation, while large models only contribute to planning.
The authors evaluate PaT across multiple models and code benchmarks , showing that it matches the accuracy of larger models (e.g., Qwen3-32B) with smaller models (e.g., Qwen3-4B) while reducing costs, and outperforms prior methods like FunCoder on complex tasks at ~60% of the computational cost. The paper’s core contributions are the PaT framework, the role division of small/large models for cost-efficiency, and empirical validation of its effectiveness on diverse code generation tasks.

**Strengths:**

1. The paper proposes the use of heterogeneous models and first trial pipeline to resolve the overhead and performance problem.
Specifically, a smaller model is first employed for reasoning—if it passes the test cases, the process terminates; otherwise, a more powerful model and code generation workflow are invoked to solve complex problems. The proposed approach follows a straightforward rationale, and the authors have conducted extensive experiments across multiple baselines to validate its effectiveness. They claim that their method outperforms the previous SOTA approach in both performance and efficiency.

2. The paper introduces a heterogeneous model framework to tackle complex problems: a more powerful large model is utilized for overall code generation planning, while a smaller model handles relatively deterministic code generation tasks. This collaborative approach aims to reduce the overall computational overhead in the code generation process.

**Weaknesses:**

1. The approach proposed in this paper is straightforward, with a key aspect being how to verify the correctness of the initial reasoning performed by the small model, which requires accurate test cases.
However, the paper states that the authors used both the built-in test cases provided with the problems and additional test cases generated by the model based on problem requirements to jointly validate the small model's outputs. This raises a concern: does using the provided test cases to verify the small model's results risk potential data leakage?

2. According to the experimental results, the proposed method demonstrates significantly and consistently superior performance compared to the previous SOTA method. While the paper claims to improve reasoning efficiency, the experiments show enhancements in both efficiency and performance over prior methods. Could the authors analyze the specific sources of these performance gains? Were the experimental settings fully aligned with those of the previous SOTA method.

3. Could the authors provide additional statistical insights—such as the proportion of problems successfully solved by the small model alone, and a comparison of token overhead between the proposed method and pre-decomposition approaches like Funcoder for complex problems—to offer a more intuitive understanding of the results?

**Questions:**

please refer to weaknesses

---

> ### Author Response · Authors · 2025-11-20
>
> ## Weakness 1
> The approach proposed in this paper is straightforward, with a key aspect being how to verify the correctness of the initial reasoning performed by the small model, which requires accurate test cases. However, the paper states that the authors used both the built-in test cases provided with the problems and additional test cases generated by the model based on problem requirements to jointly validate the small model's outputs. This raises a concern: does using the provided test cases to verify the small model's results risk potential data leakage?
>
> ---
>
> ## Answer
> We thank the reviewer for this critical question regarding data integrity. We want to clarify that no data leakage occurred. The "built-in test cases" we refer to are the public input/output examples provided within the problem description (e.g., in HumanEval), which are visible to all methods and are distinct from the hidden test suite used for final evaluation. These public examples were used only at inference time for verification and not for any model training or fine-tuning. Furthermore, our PaT policy demonstrates significant performance gains on the MBPP benchmark, which does not provide such I/O examples in its prompts. This confirms that our framework's effectiveness is not dependent on this specific verification signal and that the gains are genuine.
>
> ---
>
>
> ## Weakness 2
> According to the experimental results, the proposed method demonstrates significantly and consistently superior performance compared to the previous SOTA method. While the paper claims to improve reasoning efficiency, the experiments show enhancements in both efficiency and performance over prior methods. Could the authors analyze the specific sources of these performance gains? Were the experimental settings fully aligned with those of the previous SOTA method.
>
> ---
>
> ## Answer
> We confirm that our experimental settings were fully aligned with the previous SOTA method (FunCoder). The observed performance gains, in addition to efficiency, stem from a fundamental and critical difference in the decomposition policy.
> FunCoder employs a "Planning before Trial" approach, where the model speculatively decides whether to decompose a problem before attempting it. This policy can fail if a hard problem appears easy, causing the model to skip decomposition and fail the task.
> In contrast, our PaT policy is feedback-driven. It attempts a "Trial" first, and only after a confirmed failure does it invoke the "Plan" phase. This feedback is critical: the prompt is not "divide if you want," but a mandatory "divide because you have failed" as detailed in Figure 7. As shown in Figure 4, this means that while FunCoder may attempt decomposition, it often fails to apply it to the problems that are actually hard (which it incorrectly assessed as easy). PaT, however, reliably catches these hard problems via the failure signal and guarantees they are decomposed, leading to the superior performance.
>
>
> ---
>
>
> ## Weakness 3
> Could the authors provide additional statistical insights—such as the proportion of problems successfully solved by the small model alone, and a comparison of token overhead between the proposed method and pre-decomposition approaches like Funcoder for complex problems—to offer a more intuitive understanding of the results?
>
> ---
>
> ## Answer
> We thank the reviewer for suggesting these valuable statistical insights. We have added several key statistical analyses to Appendix A and Appendix G of the revised manuscript.
>
> Specifically, Appendix A includes the empirical validation of Theorem 3, demonstrating that the formula successfully recommends a cost-efficient model based on our experimental results. Appendix G provides additional statistics, including the proportion of problems solved by the small model alone and the detailed breakdown of our model-generated test cases.

---

### Official Review · Reviewer_kvhZ · 2025-11-01

**Soundness:** 3
**Presentation:** 3
**Contribution:** 2
**Rating:** 6
**Confidence:** 2

**Summary:**

This paper proposes PaT (Planning-after-Trial), an adaptive framework for efficient LLM-based code generation. Unlike prior Planning-before-Trial methods that plan before solving, PaT first attempts direct code generation and triggers planning only when execution fails, dynamically balancing cost and accuracy. The authors further design a heterogeneous configuration, using small models for generation and large models for planning, and provide theoretical analysis proving PaT’s cost-optimality under scaling laws. Experiments on multiple benchmarks (HumanEval, MBPP, xCodeEval) and models (Qwen3, LLaMA3, DeepSeek-Coder) show PaT achieves up to 7–8% higher Pass@1 while reducing token cost by ~40%, outperforming prior methods like FunCoder and Self-Plan.

**Strengths:**

- Novel adaptive policy: The Planning-after-Trial framework is conceptually elegant and practical, bridging efficiency and accuracy.

- Strong empirical evidence: Consistent improvements across multiple model scales and benchmarks validate robustness.-

- Theoretical rigor: Formal cost analysis and scaling-law–based proofs establish clear efficiency guarantees.

**Weaknesses:**

- Verification dependency: Performance heavily relies on the quality of automatically generated test cases. how robust is PaT under noisy or incomplete verification signals?

- Scaling analysis abstraction: The theoretical model assumes idealized uniform difficulty distribution; how does this assumption hold empirically in real-world workloads?

- Adaptivity granularity: The binary “trial vs. plan” trigger could be further refined. Could a graded or probabilistic planning policy yield smoother efficiency?

- The method can be evaluated on more recent and challenging code benchmarks such as LiveCodeBench Pro, SWEBench, etc.

**Questions:**

How does performance scale with recursion depth or number of subproblems in decomposition?

---

> ### Author Response · Authors · 2025-11-20
>
> ## Weakness 1
> Verification dependency: Performance heavily relies on the quality of automatically generated test cases. how robust is PaT under noisy or incomplete verification signals?
>
> ---
>
> ## Answer
> We thank the reviewer for this critical question, which addresses the core challenge of real-world verification. We fully acknowledge the concerns from [5, 6] and agree that model-generated test oracles are imperfect and prone to errors. To answer Question 1 directly, we did measure the "false positive" rate of our generated tests on the HumanEval benchmark and reported it in Appendix G. The analysis (detailed in Figure 8) confirms the verification signal's overall high quality. The results demonstrate that the majority of generated test suites (ranging from 55.7% to 70.3%) contain zero incorrect test cases, and when errors do occur, they are generally small in magnitude (i.e., concentrated in the 1 or 2 incorrect test categories).
>
> However, test case generation is essential because sole reliance on public tests is insufficient (lacking edge cases) or impossible (MBPP provides none). We acknowledge the possibility that external research [5, 6] may improve test fidelity, and integrating such advancements is a promising direction for future performance and efficiency gains. However, achieving a perfectly noise-free oracle is likely impossible. Therefore, a robust framework must cope with this unavoidable noise. Our "plateau heuristic" is the pragmatic heuristic designed for this purpose: it acts as an early-stopping mechanism to prevent the system from wasting computation on what are likely spurious signals, a necessary feature for managing residual imperfections.
>
>
> ---
>
>
> ## Weakness 2
> Scaling analysis abstraction: The theoretical model assumes idealized uniform difficulty distribution; how does this assumption hold empirically in real-world workloads?
>
> ---
>
> ## Answer
> The idealized uniform distribution in our theoretical model (Section 4.1) is indeed a deliberate simplification, chosen to achieve a tractable proof and establish the fundamental existence conditions for our policy. However, we argue that this assumption is deliberately conservative. As established by research on LLM scaling laws, performance does not increase linearly with model size; capability exhibits diminishing returns. This implies that real-world problem distributions are not uniform but are heavily skewed, with a large volume of "easy" tasks solvable by smaller models and a long, difficult tail of "hard" tasks that only the largest models can solve.
>
> Our uniform distribution assumption is therefore a pessimistic lower bound because it overestimates the proportion of hard tasks compared to a typical real-world workload. The fact that our theoretical analysis still proves the cost-efficiency of the PaT policy under this more challenging conservative assumption makes our findings more robust. This is further validated by our strong empirical results (Section 5) on real-world benchmarks, which do not follow a uniform distribution and where PaT demonstrates significant efficiency gains, suggesting the real-world benefits may be even greater than our conservative theory predicts.
>
> ---
>
>
> ## Weakness 3
> Adaptivity granularity: The binary “trial vs. plan” trigger could be further refined. Could a graded or probabilistic planning policy yield smoother efficiency?
>
> ---
>
> ## Answer
> We agree that a graded or probabilistic policy could potentially offer even smoother efficiency gains. The framework's true potential lies in expanding the decision space, precisely as the reviewer suggests. Future work could explore multi-level adaptive policies by using an array of models (e.g., sLM to MidLM to LLM) to achieve fine-grained efficiency gains. Furthermore, we are eager to explore dynamic multi-role systems where the high-capability LLM acts not only as the Planner but also as an available action for direct generation (LLM Generator), providing the system with a richer and more powerful set of adaptive choices.
>
> However, we chose the simplest binary "Trial vs. Plan" trigger for this work to isolate our core contribution and prove the fundamental viability and cost-efficiency of the heterogeneous architecture (sLM + LLM) itself. Using this simple binary policy allowed us to clearly attribute the efficiency gains directly to this novel architecture, which demonstrates state-of-the-art performance.

---

> > ### Author Response · Authors · 2025-11-20
> >
> > ## Weakness 4
> > The method can be evaluated on more recent and challenging code benchmarks such as LiveCodeBench Pro, SWEBench, etc.
> >
> > ---
> >
> > ## Answer
> > We thank the reviewer for this valuable suggestion. Our current work focused on rigorously validating the core cost-efficiency and adaptive principles of the PaT framework. To this end, we utilized standard foundational benchmarks (HumanEval, MBPP) and also included xCodeEval, a more complex, competition-level dataset, to demonstrate PaT's effectiveness on challenging algorithmic problems.
> >
> > We agree that SWE-Bench and LiveCodeBench Pro are crucial for testing frontier capabilities. However, these benchmarks were deliberately excluded because their inherent complexities would obscure the validation of our core algorithmic principles. In such an environment, comparing 'cost-efficiency' becomes methodologically challenging, since frontier models do not perform well. We therefore focused on benchmarks where solutions are achievable, allowing us to clearly validate PaT's core contribution to algorithmic efficiency.
> >
> > ---
> >
> > ## Question 1
> > How does performance scale with recursion depth or number of subproblems in decomposition?
> >
> > ---
> >
> > ## Answer
> > We thank the reviewer for this insightful question about how our framework scales. In our experiments, we observed that the performance scales gracefully, as the recursion depth rarely exceeded 1. This is not a limitation, but an emergent property of our framework's efficiency; the Planner (Large Model) is highly effective at creating clear, well-defined subproblems that the Generator typically succeeds on immediately, making further recursion unnecessary for the benchmarks tested. Furthermore, the number of subproblems is not a fixed hyperparameter but is dynamically determined by the Planner based on the task's complexity, as forcing a specific number was ineffective. We observed an average of 2.7 subproblems per task, which demonstrates that the decomposition is a controlled process and the associated planning cost scales linearly and manageably.

---

> > > ### Comment · Reviewer_kvhZ · 2025-11-26
> > >
> > > Thank you to the authors for the response. My concerns and questions have been largely addressed, particularly those regarding the theoretical assumptions of PaT. I will accordingly increase my confidence and lean toward acceptance of this paper.

---

> > > > ### Author Response · Authors · 2025-11-27
> > > >
> > > > We sincerely thank the reviewer for the positive re-evaluation and for confirming that our response regarding the theoretical assumptions has addressed your concerns. We remain fully available for any further discussion or questions you may have.

---

### Author Response · Authors · 2025-12-04

We sincerely thank the Area Chair and all reviewers for their time and constructive feedback.

During the discussion period, we actively worked to address the key concerns regarding theoretical validation and verification robustness. Specifically, we provided an empirical fitting of Theorem 3 to validate our model's utility, and presented a quantitative analysis of generated test cases showing that despite inherent noise, the vast majority of signals are reliable. We also significantly strengthened the manuscript by incorporating detailed comparisons with suggested references to clarify our distinct contributions.

We have decided to withdraw the paper to allow us sufficient time to fully incorporate these suggestions and further strengthen the manuscript. We appreciate the effort invested in evaluating our work.

Best regards,

Authors

---

### Note · Authors · 2025-12-04

I have read and agree with the venue's withdrawal policy on behalf of myself and my co-authors.